# Cross-feeding affects the target of resistance evolution to an antifungal drug

**Romain Durand**[1,2,3,4,5]*, **Jordan Jalbert-Ross**[1,2,3,4¤], **Anna Fijarczyk**[1,2,3,4,5], **Alexandre K. Dubé**[1,2,3,4,5], **Christian R. Landry**[1,2,3,4,5]*

**1** Département de Biochimie, de Microbiologie et de Bio-informatique, Faculté des Sciences et de Génie, Université Laval, Canada, **2** Institut de Biologie Intégrative et des Systèmes (IBIS), Université Laval, Canada, **3** PROTEO, Le regroupement québécois de recherche sur la fonction, l'ingénierie et les applications des protéines, Université Laval, Québec, Canada, **4** Centre de Recherche sur les Données Massives (CRDM), Université Laval, Québec, Canada, **5** Département de Biologie, Faculté des Sciences et de Génie, Université Laval, Québec, Canada

¤ Current address: Département de Biochimie et médecine moléculaire, Faculté de Médecine, Université de Montréal, Montreal, Canada
* romain.durand.1@ulaval.ca (RD); christian.landry@bio.ulaval.ca (CRL)

**Data Availability Statement:** All sequencing data are available at the NCBI Sequence Read Archive (SRA) under BioProject PRJNA95213: https://www.ncbi.nlm.nih.gov/bioproject/PRJNA952138. Details on demultiplexing are provided in S4 Data.

## Abstract

Pathogenic fungi are a cause of growing concern. Developing an efficient and safe antifungal is challenging because of the similar biological properties of fungal and host cells. Consequently, there is an urgent need to better understand the mechanisms underlying antifungal resistance to prolong the efficacy of current molecules. A major step in this direction would be to be able to predict or even prevent the acquisition of resistance. We leverage the power of experimental evolution to quantify the diversity of paths to resistance to the antifungal 5-fluorocytosine (5-FC), commercially known as flucytosine. We generated hundreds of independent 5-FC resistant mutants derived from two genetic backgrounds from wild isolates of *Saccharomyces cerevisiae*. Through automated pin-spotting, whole-genome and amplicon sequencing, we identified the most likely causes of resistance for most strains. Approximately a third of all resistant mutants evolved resistance through a pleiotropic drug response, a potentially novel mechanism in response to 5-FC, marked by cross-resistance to fluconazole. These cross-resistant mutants are characterized by a loss of respiration and a strong tradeoff in drug-free media. For the majority of the remaining two thirds, resistance was acquired through loss-of-function mutations in *FUR1*, which encodes an important enzyme in the metabolism of 5-FC. We describe conditions in which mutations affecting this particular step of the metabolic pathway are favored over known resistance mutations affecting a step upstream, such as the well-known target cytosine deaminase encoded by *FCY1*. This observation suggests that ecological interactions may dictate the identity of resistance hotspots.

## Author summary

Determining the paths evolution takes to make microbes resistant to antimicrobials is key to drug stewardship. Flucytosine is one of the oldest antifungals available. It is often used

All scripts and underlying data used for the analyses and to generate the figures are available at https://github.com/Landrylab/Durand_et_al_2023. The repository also contains the layouts for all assays, including a list of all strains generated in the evolution experiment. All strains, whether they are listed in one of the repository layouts or in S1 Data, are available and can be requested by email at landrylaboratory@gmail.com.

**Funding:** This work was supported by the Canadian Institutes of Health Research (CIHR) (Foundation grant 387697 to CRL) and a Genome Canada and Genome Quebec grant (grant 6569 to CRL). CRL holds a Canada Research Chair in Cellular Systems and Synthetic Biology. RD is supported by a Fonds de Recherche du Québec - Santé (FRQS) postdoctoral fellowship. AF was supported by LSARP Genome Canada (grant 10106), and an NSERC Discovery grant (RGPIN-2020-04844 to CRL). The funders had no role in study design, data collection and analysis, decision to publish, or preparation of the manuscript.

**Competing interests:** The authors have declared that no competing interests exist.

to treat cryptococcal infections. However, despite decades of use in the clinic, some details of its metabolism and of the mechanisms of resistance evolution still elude us. Flucytosine resistance is most often acquired specifically by inactivating a gene essential for the activation of this prodrug. We show that among many paths possible, one is overrepresented and involves a diversity of mutations that prevent enzyme expression or its activity. This path is preferred because these mutations also protect from the activation of the prodrug by non-mutant cells. A second, less frequent path to resistance, putatively involves a generalized response, which leads to fungal cells having an increased efflux capacity. The same mutants end up being resistant to the distinct and most widely used antifungal fluconazole. Our results show that the paths followed by evolution are influenced by microecological conditions and that resistance to unrelated drugs can emerge from the same mutations.

## Introduction

The past decades have seen great advances in the medical field, with the death rate from complications of medical and surgical care steadily declining [1]. Numbers have also gone down for some of the leading causes of death globally, notably deaths from HIV, which have decreased by 51% between 2000 and 2019 [2]. Unfortunately, some of the treatments which help increase life expectancy also lower immune defenses, which puts patients at risk of potentially life-threatening infections caused by opportunistic pathogens [3–5]. Among them, *Candida auris*, several *Candida* species and *Aspergillus fumigatus*, all drug-resistant fungal pathogens, feature on the CDC's list of antimicrobial resistance threats [6]. Among other factors, the biological similarities between fungal and host cells make it difficult to design efficient antifungals with low toxicity [7]. Consequently, there are only five main classes of antifungals [7]. One may wonder what makes a 'good' antifungal target.

The role of most antifungals is to prevent proper synthesis of the membrane or cell wall, because they are made of components not found in mammalian cells (ergosterol, chitin-glucan, for example). If we were to look for other molecules, should the antifungal target an essential metabolic end product? An enzyme in a metabolic pathway? If the enzyme catalyzes an early step of a specific pathway, resistance could potentially arise through loss of function downstream, for instance when a toxic molecule is produced as a byproduct or through the conversion of a prodrug. Among the considerations for the development of powerful drugs targeting cell metabolism is the organization of cell communities *in vivo*. For instance, treatment can have drastically different outcomes depending on the microbe lifestyle, fungi in biofilms being notoriously less susceptible to antifungals [8].

Interestingly, despite the urgent need for better antifungals, the mechanisms of resistance are still poorly understood outside of the direct drug targets [3]. Particularly in the medical field, it is common practice to sequence as few as a single gene to look for 'common' resistance-conferring mutations, which introduces a bias in databases such as MARDy [9]. In doing so, resistance is too often studied through a keyhole. Even more challenging is deciphering if the mechanisms of resistance to a specific drug are general, or if they depend on the genetic background or the environmental conditions in which they take place. Here, we leveraged the power of experimental evolution coupled with whole-genome sequencing (WGS) to capture a large number of mutations conferring resistance to 5-fluorocytosine (5-FC, also known as flucytosine) using the model yeast *Saccharomyces cerevisiae*. We focused on 5-FC because it is among the oldest antifungals, it is listed by the WHO as an essential medicine

[10], and its metabolism is documented enough that we can interpret this data with respect to the adaptive mechanisms and rationalize the paths to resistance [11].

5-FC is a prodrug that is not toxic to fungi before it is metabolized [12]. Its metabolism involves several steps, which ultimately lead to cell toxicity (Fig 1). First, it is imported into the

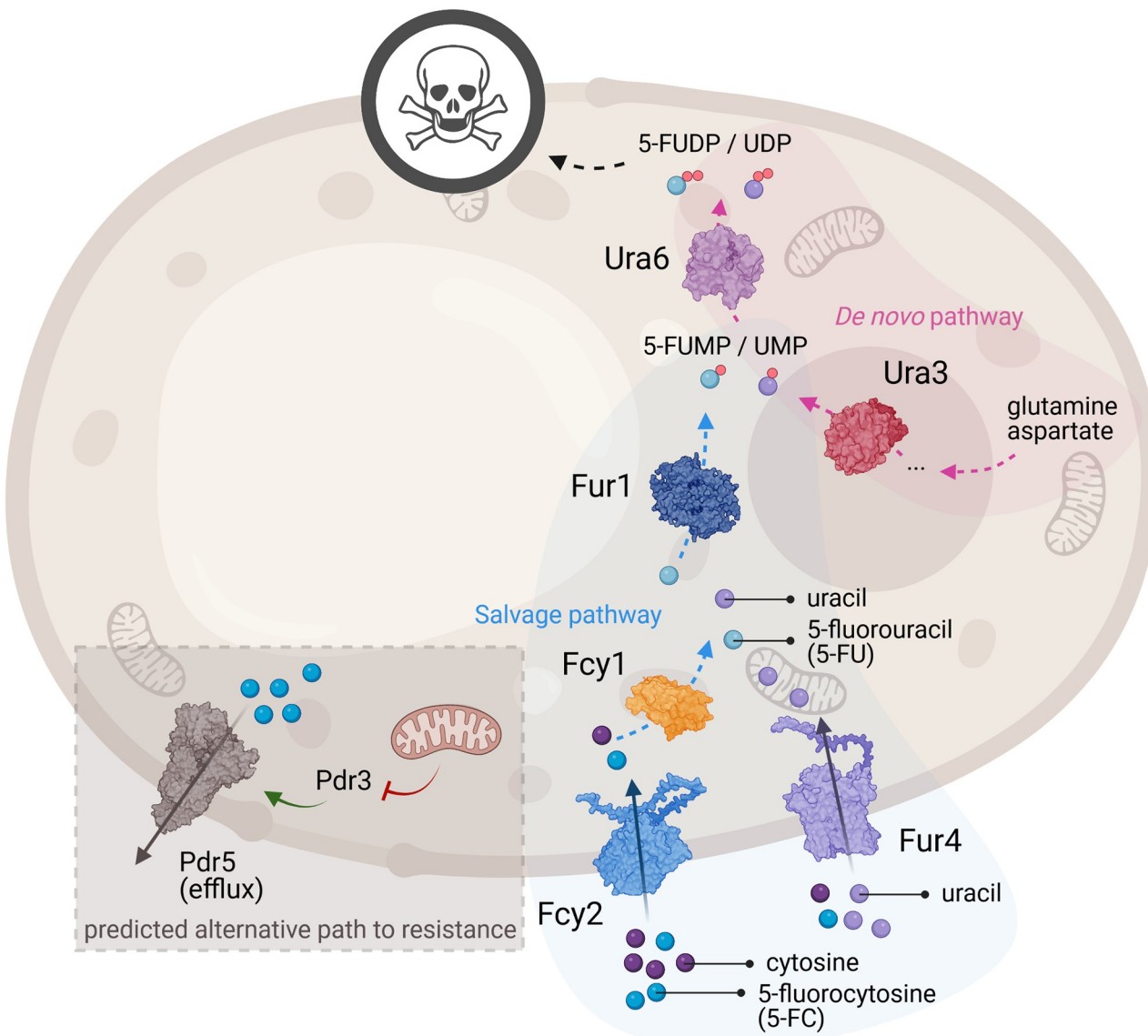

**Fig 1. Pyrimidine and 5-FC import and metabolism around some of the known and potential mechanisms of resistance.** 5-Fluorocytosine (5-FC), cytosine, uracil and 5-fluorouracil (5-FU) are represented as blue, dark purple, light purple and light blue beads, respectively. For clarity, only two permeases known to import pyrimidines are shown: Fcy2 and Fur4. In prototrophic yeast strains, pyrimidines can be synthesized *de novo* from amino acids such as glutamine and aspartate [29]. The steps leading to the obtention of the nucleotide precursors uridine monophosphate (UMP) and uridine diphosphate (UDP) are catalyzed by Ura3 and Ura6, respectively, in the *de novo* pathway [30,31]. Alternatively, pyrimidines can be imported from the medium [32]. Uracil can also be obtained by deamination of cytosine by Fcy1 [33]. Finally, UMP can be obtained by the addition of a phosphate group by Fur1 [34]. 5-FC is a prodrug, which is metabolized the same way cytosine is, except its deamination leads to the obtention of the cytotoxic compound 5-FU [35]. Based on work done in laboratory strains or other fungal species, resistance to 5-FC can arise when either of the following three steps are compromised: import, conversion and activation, typically through mutations inactivating Fcy2 (for example), Fcy1 and Fur1, respectively [36]. We also hypothesize an alternative path to resistance in the pleiotropic drug response, wherein loss of mitochondrial function leads to derepression of the transcription factor Pdr3, which in turn overexpresses efflux pumps such as Pdr5. Figure created with BioRender, using the structures of Fcy1 (1P6O), Fur1 from *C. albicans* (7RH8), Ura3 (3GDL), Ura6 (1UKZ) and Pdr5 (7P04) and AlphaFold predictions of Fcy2 (P17064) and Fur4 (P05316) [37,38].

cell by several permeases, typically those involved in the import of exogenous pyrimidines (Fcy2, Fur4, Fcy21, Fcy22) [13]. Next, 5-FC is deaminated by the cytosine deaminase Fcy1 into the cytotoxic compound 5-fluorouracil (5-FU). 5-FU is a molecule known outside of the field of antifungals that has been used to treat cancers for several decades [14]. Its main activation relies on the addition of a phosphate group by the uracil phosphoribosyltransferase Fur1. Incorporation of fluoronucleotides into RNA and DNA, as well as inhibition of the thymidylate synthase either cause growth arrest or cell death [14–16]. Resistance to 5-FC is typically conferred by mutations in the genes involved in its metabolism, preventing its import (*FCY2*), its conversion into 5-FU (*FCY1*) or the activation of 5-FU (*FUR1*) (Fig 1). *FCY* genes were named after their phenotypes associated with FluoroCYtosine resistance. Mutations in *FCY1* have regularly been reported as being associated with 5-FC resistance in clinical isolates of a broad range of pathogens such as *Candida albicans*, *Candida dubliniensis*, *Nakaseomyces* (*Candida*) *glabrata* and *Clavispora* (*Candida*) *lusitaniae* [17–20].

Few alternative paths to 5-FC resistance have been shown, with some studies pointing at a putative pleiotropic drug response resulting in efflux-mediated resistance (Fig 1) [21–25]. This mechanism is far better characterized for azoles. Azole treatment typically favors the formation of $rho^0$ mutants (also known as *petite* mutants), which are characterized by their loss of mitochondrial DNA. This results in an alleviation of repression of the transcription factors (Pdr1, Pdr3) regulating the expression of efflux pumps [26]. Some of these efflux pumps, for example Pdr5, end up overexpressed, conferring cross-resistance to several antifungals [26].

Because of the tight relationship with pyrimidine metabolism, the use of auxotrophic laboratory yeast strains to study the mechanisms of resistance may paint a distorted picture of how resistance to 5-FC and 5-FU comes to evolve outside the lab. Notably, it has been shown that the presence of uracil in the growth medium triggers a negative feedback by repressing the expression of the uracil permease Fur4 [27], potentially limiting cell entry by 5-FC. Not only does the use of specific lab strains narrow the number of detectable paths to 5-FC resistance, it also prevents the detection of potential epistatic effects. Indeed, the impact of the genetic background on the evolution of resistance is still poorly understood and has only recently become one of many interesting avenues of research in the field [28]. Finally, searching a relatively small gene space for resistance-conferring mutations often limits interpretations. With the decreasing costs of WGS, we can now hope to reach saturation, in that sequencing a larger number of mutants would not substantially increase the number of relevant mutations.

For the aforementioned reasons, we set up to generate hundreds of 5-FC resistant mutants, subjecting two wild prototrophic strains of *Saccharomyces cerevisiae* to experimental evolution in minimal medium. Their relative fitness was estimated in several conditions to test for respiration capacity, fitness cost and cross-resistance. We show that about a third of all mutants developed generalized resistance through loss of mitochondrial function. The majority of the remaining two thirds evolved resistance to 5-FC and 5-FU. To explain their phenotype, the genomes of 276 of these mutants were analyzed by WGS. In particular, we sought out to determine if the resistance phenotype conferred by a given mutation was dependent on the genetic background, or if there were genetic interactions for a given background. For both backgrounds, virtually all sequenced mutants evolved resistance through loss-of-function mutations in the gene *FUR1*. Interestingly, none of the sequenced genomes showed any mutation in the well-known resistance gene *FCY1*, which we show is the result of negative selection imposed by the 5-FU-producing *FUR1* mutants. Ultimately, sequencing a large number of genomes also led to the identification of rare alleles most likely sufficient to confer resistance to 5-FC.

## Results

### Mutants evolved in 5-FC develop cross-resistance to 5-FU

Two environmental strains of *S. cerevisiae*, LL13-040 and NC-02 were subjected to parallel experimental evolution, which generated 682 5-FC resistant mutants. All candidate resistant strains were streaked and spotted on YPG agar medium to evaluate mitochondrial function. *rho*⁻ mutants amounted to 83/296 (28%) and 140/386 (36%) for LL13-040 and NC-02, respectively. The fitness of 408 5-FC resistant mutants was estimated by measuring colony growth through time on solid media.

For both backgrounds, most mutants show an expected increase in relative fitness (relative to the parental strain) in minimal medium supplemented with 5-FC compared to control conditions (rich or minimal medium) (Fig 2). *rho*⁻ mutants show a lower relative fitness in drug-free media, with LL13-040 *rho*⁻ mutants displaying a larger fitness defect than NC-02 counterparts. For both backgrounds, the relative fitness in minimal medium, the same medium in which the strains were evolved, is generally lower or equivalent to that of the parental strains. Additionally, when we performed the evolution experiment in the absence of 5-FC, no resistant mutant was generated. Both observations suggest that, if mutants evolved to grow better in minimal medium, none of the evolutionary routes coincidentally conferred resistance to the antifungal. We also measured the growth on minimal medium supplemented with 5-FU, the converted form of 5-FC obtained by Fcy1-mediated deamination in the pyrimidine salvage pathway (Fig 1). *rho*⁻ and *rho*⁺ mutants show a similar relative fitness, which is about twice that of the parental strains in minimal medium supplemented with either 5-FC or 5-FU.

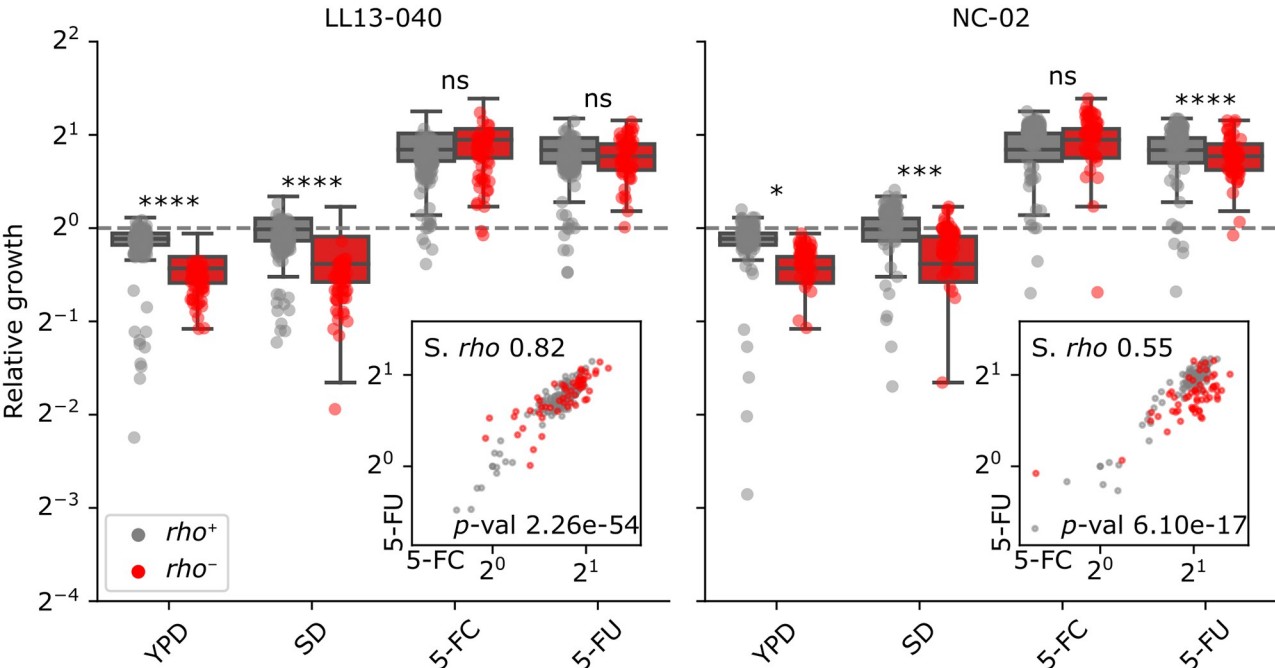

**Fig 2. Growth of individual mutants relative to the parental strains in various media.** Relative growth corresponds to the mean area under the curve (AUC, calculated on 22 h) from four replicate colonies, normalized by the WT for individual strains arrayed on solid media: YPD, SD, SD + 25 μg/mL 5-FC and SD + 6.25 μg/mL 5-FU. A two-way ANOVA followed by Tukey's multiple comparison test was performed to compare the relative fitness of *rho*⁺ mutants to that of *rho*⁻ mutants for each background in each condition. Statistical significance is as follows: ****, adj. *p*-value < 0.0001; ***, adj. *p*-value < 0.001; *, adj. *p*-value < 0.05; ns, not significant. Insets show the correlation between relative fitness in SD + 5-FC and relative fitness in SD + 5-FU, with the corresponding Spearman's rank correlation coefficient (S. *rho*) and *p*-value.

Interestingly, the relative fitness in 5-FC significantly correlates with the one in 5-FU, hinting at a mechanism of resistance that would take place downstream of the conversion of 5-FC into 5-FU or that affects intake of these molecules. We hypothesized this mechanism to be either the pleiotropic drug response in the case of *rho*⁻ mutants, or the inactivation of *FUR1* in the case of *rho*⁺ mutants.

## *rho*⁻ mutants are cross-resistant to fluconazole

To test if loss of respiration induces a pleiotropic drug response, we first evaluated the fitness of *rho*⁻ mutants in rich medium, as well as in minimal medium, supplemented or not with several antifungals belonging to distinct classes. We tested growth in the presence of three other classes of clinical antifungals: echinocandins (micafungin and caspofungin), polyene (nystatin) and azole (fluconazole). Because of the slow growth of *rho*⁻ mutants in some antifungal conditions, we incubated the plates at 37˚C. In these conditions, *rho*⁻ mutants display a higher relative fitness in 5-FC than *rho*⁺ mutants, while still showing a fitness defect in control media (Fig 3A). Interestingly, *rho*⁻ mutants of both backgrounds show cross-resistance to fluconazole, with a relative growth more than twice that of the parental strain for NC-02 (Fig 3A). Also noteworthy is their fitness in micafungin, although this is likely the result of a very low growth rate (S1 Fig).

Since the loss of mitochondrial function in *rho*⁻ mutants has been shown to lead to generalized resistance, we hypothesized that cross-resistance to 5-FC and fluconazole results from an increased expression or activity of efflux pumps. To test this hypothesis, we performed a rhodamine retention assay. Rhodamine is a known substrate of ABC transporters and its accumulation in cells has been shown to be inversely correlated with efflux capacity [39]. We find that virtually all *rho*⁻ mutants display a lower count of rhodamine-fluorescent cells compared to their parental strain, confirming they have gained an increased efflux capacity (Fig 3B). Notably, both parental strains accumulate much less rhodamine than a laboratory strain deleted for all ABC transporters, meaning that parental strains were already capable of rhodamine efflux in the conditions of the experiment. Even though the NC-02 strain appears to have a diminished efflux capacity compared to the LL13-040 strain, the *rho*⁻ mutants of both backgrounds display comparable rhodamine accumulation levels (Fig 3B). Therefore, the net gain of efflux capacity for NC-02 mutants is higher than that of LL13-040 counterparts. This efflux capacity is correlated with the measured growth in fluconazole, however the correlation is statistically significant only for NC-02 (Fig 3C). For both backgrounds, the correlation with the measured growth in 5-FC is not significant (Fig 3D). This could be explained by a lack of statistical power, but in all likelihood, a complex regulation drives the expression of multiple transporters involved in resistance to fluconazole, as well as 5-FC. Another potential mechanism is that resistance to 5-FC could be binary, whereby enough efflux above a threshold would confer resistance, whereas it would be gradual for fluconazole, creating a correlation between efflux and growth.

## *rho*⁺ mutants display a large number of distinct loss-of-function mutations in *FUR1*

Next, we focused on the mechanisms of resistance at play in *rho*⁺ mutants. To make sure we identified resistance conferring mutations, including potentially rare ones, we sequenced the genomes of most assayed *rho*⁺ mutants (149/160 LL13-040 mutants and 127/134 NC-02 mutants). Overall average sequencing coverage was around 60X, with no apparent CNVs (S2 Fig). Two computational methods were carried out to detect indels and SNPs, one with stringent criteria ("samtools"), and one with more tolerant criteria ("gatk"). Because of the

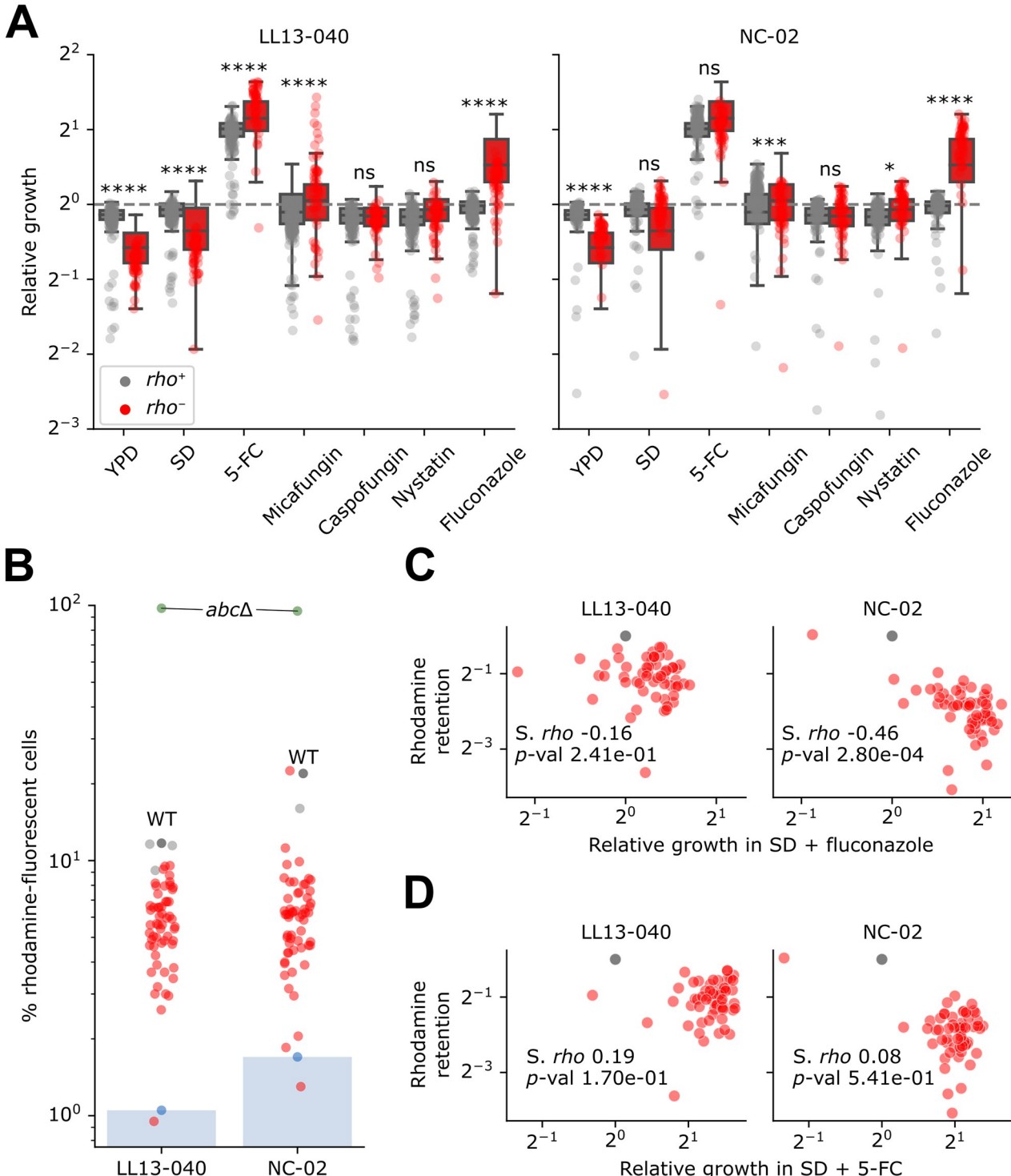

**Fig 3. Cross-resistance of *rho⁻* mutants to 5-FC and fluconazole may be explained by efflux capacity.** A) Relative growth corresponds to the mean area under the curve (AUC, calculated for the first 22 h of incubation) from four replicate colonies, normalized by the WT for individual strains arrayed on solid media: YPD, SD, SD + 25 μg/mL 5-FC, SD + 0.5 μg/mL micafungin, SD + 2 μg/mL caspofungin, SD + 16 μg/mL nystatin and SD + 64 μg/mL fluconazole. A two-way ANOVA followed by Tukey's multiple comparison test was performed to compare the relative fitness of *rho⁺* mutants to that of *rho⁻* mutants for each background in each condition. Statistical significance is as follows: \*\*\*\*, adj. *p*-value < 0.0001; \*\*\*, adj. *p*-value < 0.001; \*, adj. *p*-value < 0.05; ns, not significant. B) Rhodamine retention assay. Blue bars indicate the resulting detection thresholds (one negative control for each background i.e. cells not treated with rhodamine). *abc*Δ (s_012 in S1 Data) was used as the positive control. C, D) Rhodamine retention (data from panel B normalized with the WT) and relative growth in C) fluconazole or D) 5-FC (data from panel A) are compared, with the corresponding Spearman's rank correlation coefficient (S. *rho*) and *p*-value.

frequency of mutations observed in *FUR1* in our preliminary analysis, we also directly amplified and Sanger sequenced *FUR1* (including the promoter region) from the same genomic DNA for most of these mutants. We included WT controls as well as *fcy1Δ* and *fcy2Δ* mutants for all three analyses (samtools, gatk and Sanger).

Supporting our hypothesis, most sequenced genomes (258 out of 276 mutants) harbored a mutation in *FUR1* (Fig 4A).

Aggregating results from samtools, gatk and Sanger sequencing, we identified 118 distinct SNPs or indels in *FUR1* (S2 Data). The use of two computational methods to call variants proved relevant, since part of the variants were detected by only one method (Fig 4A). As expected, gatk, with its set of more permissive criteria, detected variants in around 500 genes in each background, whereas samtools detected variants in only 150 genes on average (Fig 4A). Variants in other genes than *FUR1* were rarely detected by both computational methods, which might be linked to the quality of their call (Fig 4A). Overall, approximately 120 genes per individual genome were found to have variants by either method (Fig 4B). This large number is a result of the filtering criteria, which were set to be more flexible to make sure rare mutations would not be discarded. Ultimately, the number of unique variants found in a given gene appeared as a good indicator to sort through the noise. Specifically, the number of unique variants found in *FUR1* was much larger than in any other gene, which is strong evidence that *FUR1* constitutes a hotspot for 5-FC resistance-conferring mutations (Fig 4A).

Out of all strains for which *FUR1* was Sanger sequenced (n = 268), all six controls (parental strains, *fcy1Δ* and *fcy2Δ* mutants) and only 13 mutants carried the WT allele of *FUR1*. Through a detailed look into our dataset, we found three candidate target genes which could alone explain the resistance phenotype of these 13 strains carrying a WT *FUR1* sequence: *URA6*, *GFA1* and *ARG2*. Two LL13-040 strains and one NC-02 strain carried a background-specific mutation in *URA6*: R52S for the former, G73S for the latter. Eight NC-02 strains carried one of four mutations in *GFA1*: V478F, G482R, G492S and N496K. One NC-02 strain carried a K113* mutation in *ARG2*. Interestingly, the corresponding mutants display a relative fitness distinct to that of *FUR1* mutants (Fig 4C and S3 Data). Notably, the mutations detected in *URA6* and *GFA1* are harbored by strains resistant to 5-FU, whereas the *ARG2* mutant is only mildly resistant to 5-FC in our experimental conditions (Fig 4C). All identified mutations in these three genes (n = 7) are predicted by mutfunc (see methods section) to impact conserved residues with five out of seven also predicted to impact protein stability. Their effect is therefore likely to be through loss of function.

Among the sequenced strains for which no clear candidate resistance mutation was identified, one appears to have been initially miscategorized as resistant to 5-FC, four display the same relative fitness in 5-FC and 5-FU as *FUR1* mutants (and are therefore likely *FUR1* mutants as well) and one appears to be mildly resistant to 5-FC but not 5-FU (Fig 4C). Finally, most unsequenced *rho*+ mutants showed little to no resistance, which suggests they were false positives of the evolution experiment (Fig 4C).

To further investigate the extent and potential mechanisms of resistance, 35 resistant isolates (across both backgrounds) for which mutations were predicted in *FUR1* were tested in a growth assay, alongside the parental strains, and the deletion mutants for *FCY1*, *FCY2* and *FUR1* (s_004, s_005, s_006, s_009, s_010, s_011 in S1 Data). All tested mutants showed a relative fitness similar to their null mutant counterpart (Fig 4D). This suggests most mutations inactivate the function of *FUR1* and are sufficient to explain the resistance phenotypes, meaning that mutations in other genes likely play no role (Fig 4B). Importantly, the relative fitness of the *fcy1Δ* mutants was comparable to that of all tested *FUR1* mutants. The reason why inactivating mutations in *FCY1* were not picked up in our experiment is thus not because they provide lower resistance levels than *FUR1* inactivating mutations in these conditions (see below).

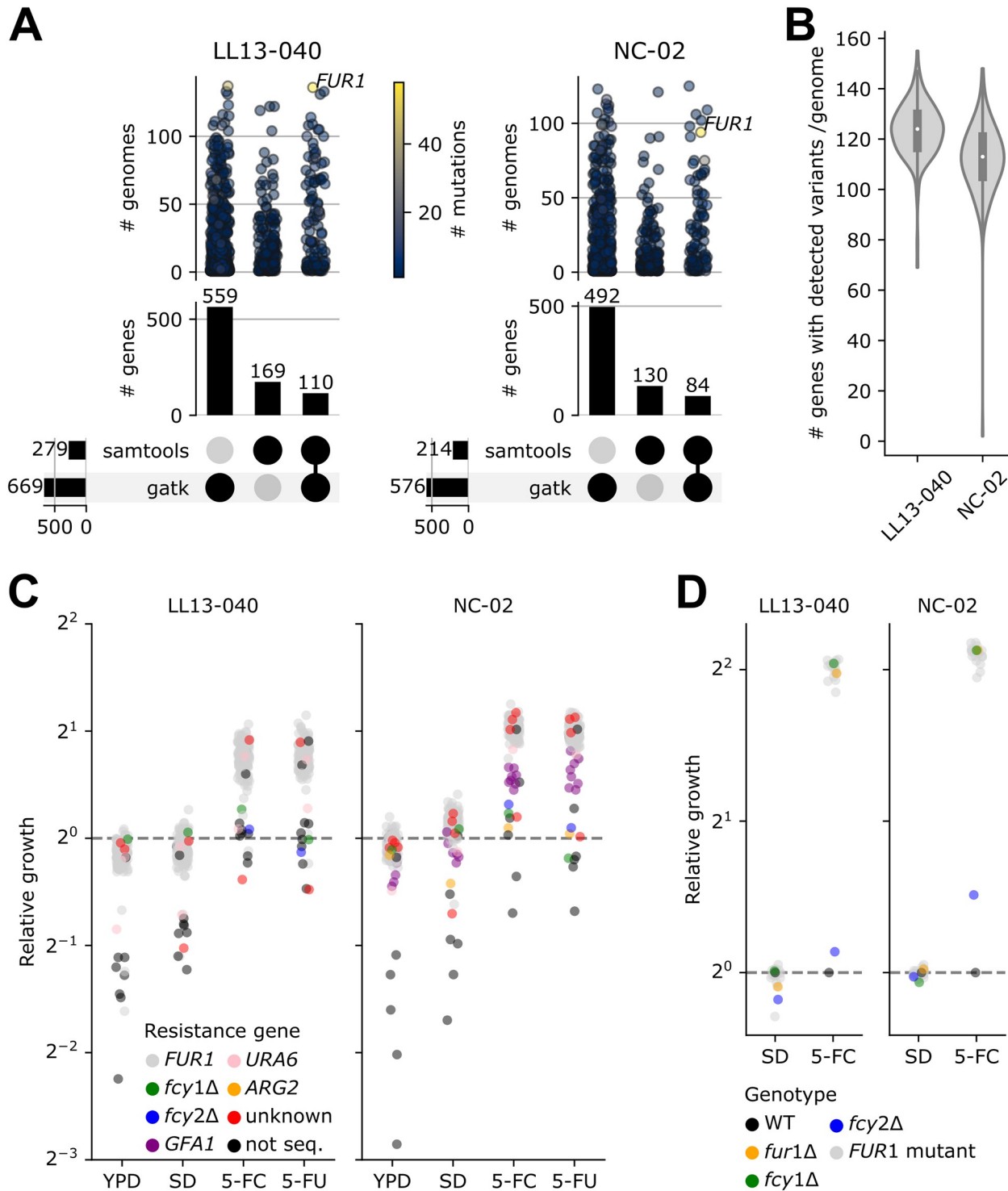

**Fig 4. Genome sequences analysis and validation.** A) Upset plots showing the number of unique mutated genes (# genes) detected by gatk, samtools or both for each background. Categorical plots on top show the number of distinct genomes for which at least one variant was detected in a given gene (one dot per gene, colored according to the number of distinct mutations which have been detected). B) Distribution of the number of genes with variants per genome (as detected either by gatk or samtools) for each background. C) Data from Fig 2, where dots are colored by the mutated/deleted gene most likely to confer resistance. D) Growth assay in liquid medium for 35 resistant mutants for which a variant in *FUR1* was detected. Cultures were inoculated in SD with or without 100 µg/mL 5-FC. Relative growth corresponds to the mean area under the curve from two replicates (AUC, calculated on 46h) normalized by the WT.

Most mutations identified in *FUR1* were confirmed by our three detection methods (samtools not picking up indels), with all samples sequenced by Sanger (268 out of 282 samples sequenced by WGS) confirming what had been detected by WGS (Fig 5A). One mutation (R110G) stands out as being much more prevalent than others in LL13-040 genomes (Fig 5A). Further examination confirmed that it is shared among 39 strains which arose from the same preculture (S3 Fig). This could be explained by a rapid fixation of the mutation that was segregating in the preculture. It is important to note however that it constitutes a rare case, since we often selected less than 20 mutants per preculture, which generally carried different mutations each (S3 Fig).

To understand why all validation mutants behaved like the null mutant, we compared their predicted impact to that of all possible substitutions. Out of 4,104 possible missense mutations in Fur1, 1,279 are predicted to impact stability. Our dataset captured a total of 76 missense mutations, 42 of which are predicted to impact stability. This is significantly more than what can be expected by chance (Fisher's exact test *p*-value = 1.39e-5, S4A Fig). The same observation can be made of mutations predicted to impact function based on conservation, with 67 out of 2,667 being captured in our dataset (Fisher's exact test *p*-value = 4.96e-6, S4B Fig). The results are consistent with loss-of-function mutations. We find that the substitutions in our dataset are predicted to destabilize Fur1 with generally high positive ΔΔG values, with no apparent bias in terms of position along the sequence (S4A and S5 Figs). Expression of *FUR1* from a centromeric plasmid under its native promoter restored sensitivity in most tested mutants (n = 32), including all with a mutation introducing a stop codon, confirming the detected mutations lead to loss of function (Fig 5B). Interestingly, some mutations could only be partially complemented, with four strains growing at more than 75% the rate of their corresponding control (vector with no expressed ORF, Fig 5B). To better understand this dominant negative effect, all residues for which we detected a mutation in our dataset were mapped on the predicted structure of the Fur1 tetramer for which the *C. albicans* structure is available (PDB: 7RH8). We find that most mutated residues are not located close to the active site, but instead appear to be involved in structurally important folds and helices (Fig 5C). Additionally, some mutated residues reside at the interface between two chains, which could indicate that the mutation prevents proper assembly of the tetramer (Fig 5C). Namely, G104 and G205 appear to interact each with a cysteine residue from the adjacent chain (Fig 5C insets). This observation could explain why the mutations G104D and G205W cannot be complemented with the WT allele of *FUR1* as they could also perturb the assembly of tetramers that would include WT chains.

## *FUR1* path to resistance prevents selection of *FCY1* mutations

One of the known resistance genes to 5-FC encodes for the cytosine deaminase Fcy1, which acts upstream of Fur1 in the pyrimidine salvage pathway (Fig 1). We included *fcy1Δ* mutants in our validation experiments as positive controls. However, we were surprised to find no single Fcy1 mutants in our set of sequenced genomes. As loss of function in *FCY1* is sufficient to lead to resistance and many mutations in the gene are destabilizing enough to cause resistance [41], one would expect *FCY1* mutations to also be frequent in our experiments.

To validate the absence of detectable mutations in *FCY1*, we randomly selected mutants (n = 14) and performed Sanger sequencing: all carried the wild-type sequence. Next, we confirmed that *FCY1* is well expressed in LL13-040, using a mEGFP protein fusion (S6A Fig). Finally, we evaluated the functionality of Fcy1 in parental strains. A growth assay using cytosine as the sole nitrogen source in minimal medium confirmed that both strains can grow, which would not be possible without the deamination of cytosine performed by Fcy1 (S6B Fig).

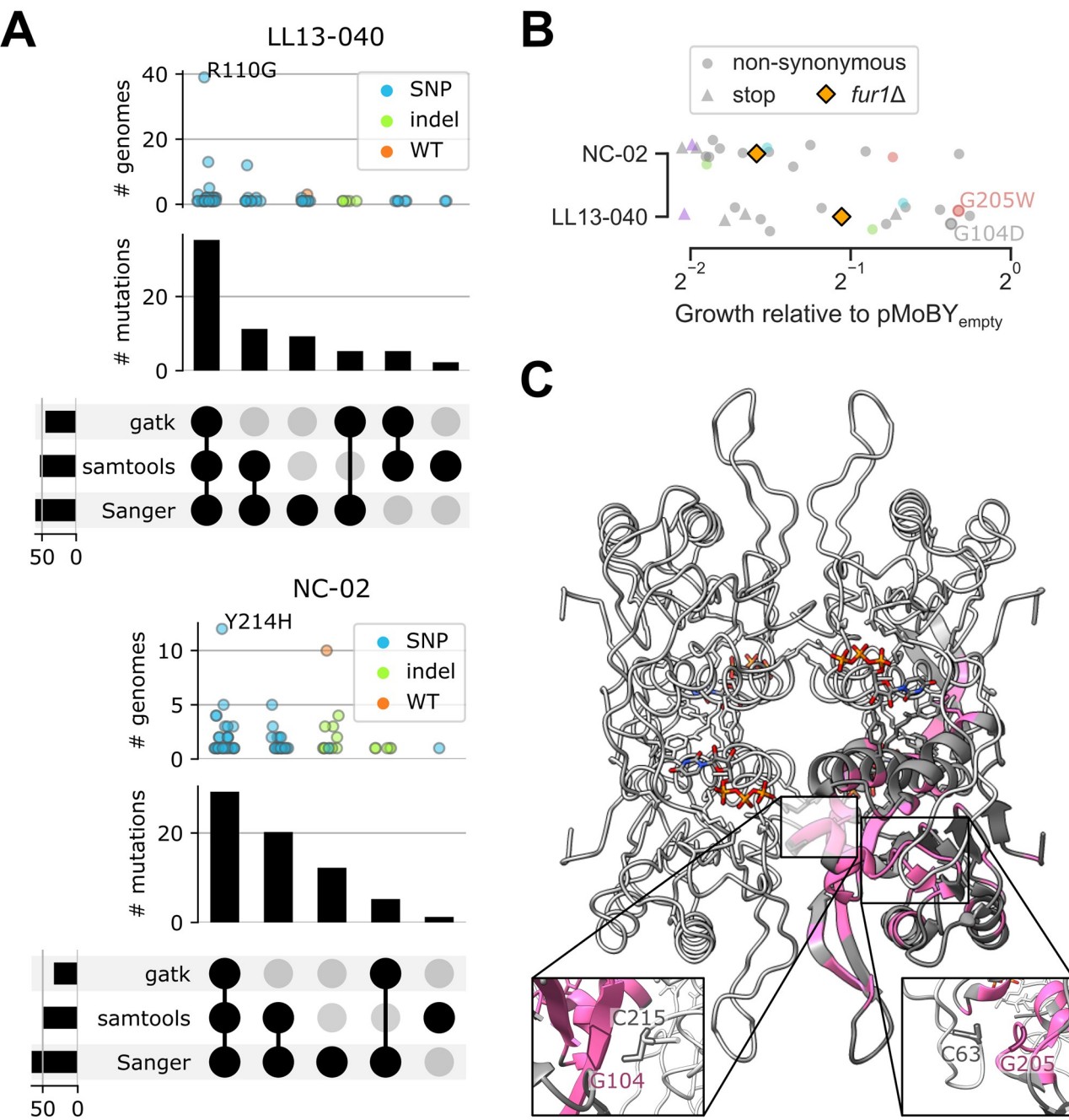

**Fig 5. Mutations observed in Fur1 lead to loss of function.** (A) Upset plots showing the number of unique mutations identified in Fur1 for each background. Categorical plots on top show the number of distinct genomes for which the same mutation was detected. Only one amino acid change resulted from two adjacent mutations (back-to-back in the same codon), all others corresponding to a single SNP or indel. B) Growth assay for 32 out of 35 resistant mutants tested in Fig 4D, with pMoBY expressing or not *FUR1* from its native promoter. Relative growth corresponds to the area under the curve calculated on 25 h for the mutant with pMoBY-*FUR1* divided by the same parameter for the mutant with pMoBY. Mutants were grown from single colonies in SD + 100 µg/mL 5-FC (+ G418 to maintain pMoBY). Colored dots indicate the same mutation was detected in the mutants from both backgrounds. Two mutations for which expression of the WT allele of *FUR1* did not restore 5-FC sensitivity are highlighted. A closer look at the residues in the next panel suggests a putative role in assembly of the tetramer. C) Location of substitutions from our dataset on the predicted structure of Fur1, visualized using UCSF ChimeraX [40]. The tetramer (*C. albicans* Fur1 assembly 7RH8) is represented in light gray, with 1 chain in dark gray. Residues found to be mutated at least once in our dataset are colored in pink. UTP molecules are represented in sticks colored by atom. Two insets zoom in on one interface each (the slightly opaque one depicts the interface at the back of the represented structure). The highlighted residues (each one from a different monomer) appear to interact with one another to ensure proper assembly of the tetramer.

Next, we sought out to confirm the fitness of the *fcy1Δ* mutants in liquid medium compared to our 5-FC mutants as a lower fitness could explain why they were not picked up in our experiments. We performed growth assays to calculate the relative fitness in three conditions with minimal medium: with or without 5-FC and one condition where the only nitrogen source is cytosine. As expected, the *fcy1Δ* mutants fared well compared to our evolved mutants in 5-FC but displayed the lowest relative fitness among *rho*+ mutants in cytosine (S7A Fig). In comparison, virtually all *rho*+ mutants have little to no fitness tradeoff in cytosine, indicating that *FCY1* is functional.

Interestingly, when we compared the relative fitness measured in liquid medium versus the one measured on solid medium, we observed that the *fcy1Δ* mutant shows a significant fitness defect on solid medium when grown alongside other mutants (S7B Fig). A recent study by our group showed that *fcy1Δ* could grow on plates supplemented with cytosine as the only nitrogen source as long as strains with a functional *FCY1* allele (*FCY1*+) were growing in proximity [41]. The auxotrophy of a *fcy1Δ* strain can therefore be rescued by cross-feeding uracil from *FCY1*+ strains. We hypothesized that this could also occur for 5-FU and, through interactions between cells of different genotypes and diffusion in the medium, it would lead to strains with a loss of function of *FUR1* (*fur1*- strains) taking over strains with a loss of function of *FCY1* (*fcy1*- strains) during experimental evolution.

Specifically, if a mutant is sensitive to a compound effluxed by its neighbor, its growth will be inhibited. The ultimate toxic compound produced from the prodrug is 5-FUMP, downstream of Fcy1 and Fur1. This means that in a complex community, a *FCY1*+ strain could produce 5-FU, which would inhibit its growth and that of *fcy1*- neighboring cells too, provided they are *FUR1*+ (Fig 6A). On the other hand, a *fur1*- strain would be resistant to its own production of 5-FU and to that of neighboring *FCY1*+ cells (Fig 6A). During experimental evolution, both *fcy1*- and *fur1*- mutants could arise but because they do so at low frequency at first, only the latter would increase in frequency. Pure cultures of *fcy1*- and *fur1*- would however be both equally resistant to 5-FC, as our previous results revealed. If this model is correct, conditioning the medium with a *fcy1Δ* strain should allow growth of a *fur1Δ* strain, whereas the opposite would lead to growth inhibition. We confirmed this prediction (Fig 6B). For both backgrounds, conditioning the medium with the *fcy1Δ* mutant led to the *fur1Δ* mutant growing significantly better than the WT. On the other hand, conditioning the medium with the *fur1Δ* mutant led to growth inhibition of both the WT and the *fcy1Δ* mutant.

In this assay we show the ultimate outcome of applying selective pressure to both mutants, but what happens if the only source of 5-FC comes from the solid medium on which the population is plated? To evaluate such a scenario, we designed a competition experiment, in which both mutants are grown individually with no selective pressure. For each background, the *fcy1Δ* is pooled with the *fur1Δ* culture. Serial dilutions are then plated on large Petri dishes with 5-FC, and mutants are identified by multiplexed colony PCR. We show that the survival of *fcy1Δ* mutants significantly depends on the inoculum concentration (Fig 6C). Specifically, if a 1:1 mix of *fcy1Δ* and *fur1Δ* mutants is plated at $10^{-5}$ OD$_{600}$, the same ratio will be observed for the growing colonies. However, when the same mix is plated at a concentration 100 times higher, all tested colonies are *fur1Δ*, meaning *fcy1Δ* cells are killed by 5-FU cross-feeding. To further confirm the link between cell density and cross-feeding, we performed an inhibition assay, where a culture of the *fur1Δ* mutant is spotted on a *fcy1Δ* lawn spread on medium with 5-FC. For both backgrounds, we were able to see a clear inhibition zone around the *fur1Δ* spot (Fig 6D).

In summary, the design of our experimental evolution, where the mutants are selected by spotting precultures on a plate, negatively selects against spontaneous *FCY1* mutants. This would likely occur as well in a mixed community.

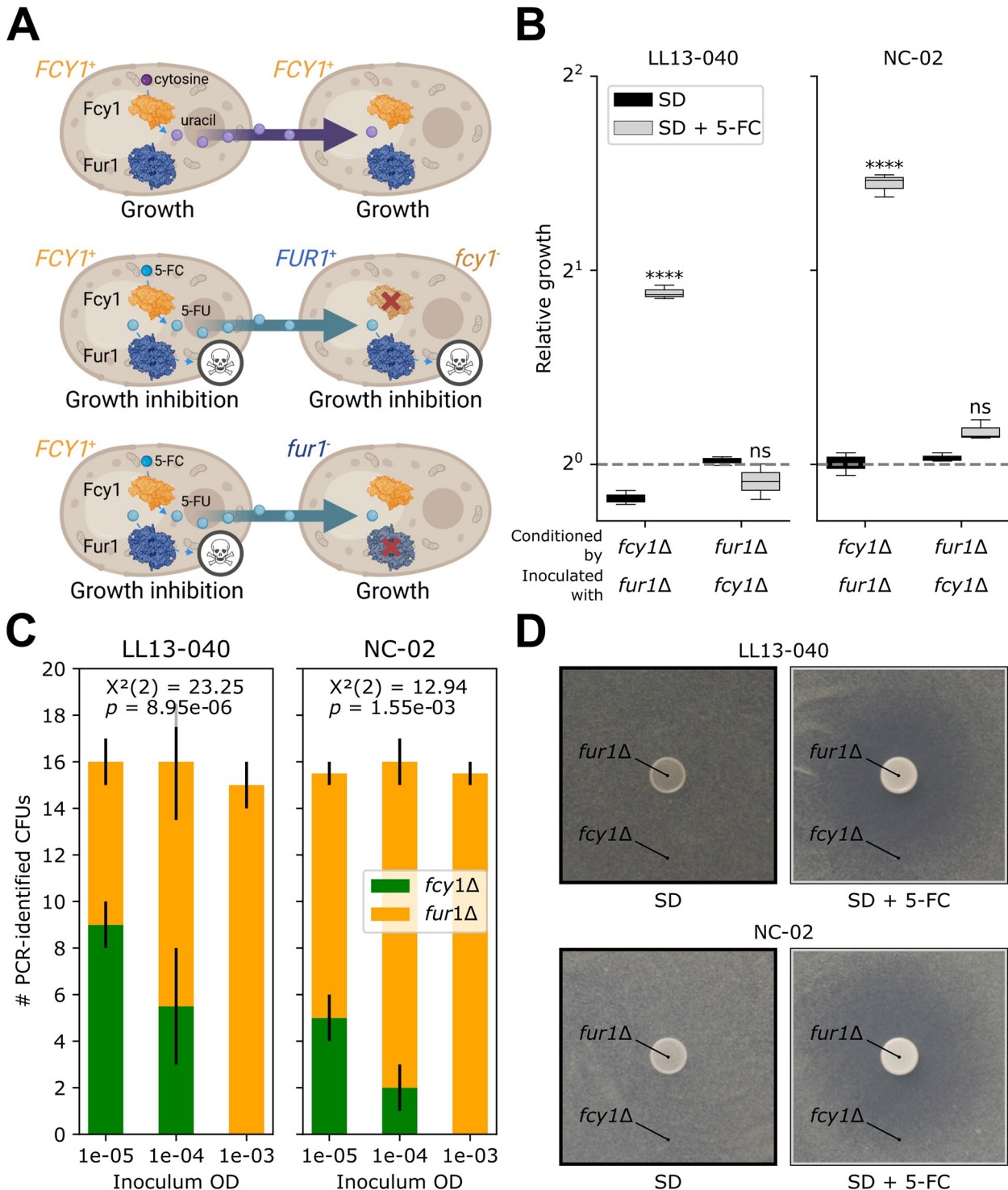

**Fig 6. *FCY1*-mediated resistance is context-dependent.** A) Predicted cross-feeding interactions between cells with different alleles of *FCY1* and *FUR1*. Uracil (top) and 5-FU (middle) are expected to diffuse in the medium and be converted by Fur1 of neighboring cells, which would allow or inhibit growth, respectively. Notably, the growth of any *FUR1*⁺ cells, regardless of their *FCY1* allele, will be inhibited because of 5-FU cross-feeding. On the contrary, any *fur1*⁻ cells would grow despite 5-FU cross-feeding (bottom). Figure created with BioRender, using the structures of Fcy1 (1P6O) and Fur1 from *C. albicans* (7RH8). B) Medium conditioning assay. First, cultures of the deletion mutants were prepared in SD with or without 1.56 μg/mL 5-FC. After reaching an $OD_{600}$ of 0.6, the cultures were filtered and the medium was transferred to a 96-well plate. Cultures were inoculated with either the WT or the other deletion mutant. Relative growth corresponds to the area under the curve (AUC) normalized by the WT and was obtained from biological triplicates. A two-way ANOVA followed by Tukey's multiple comparison test was performed to compare the

relative growth in SD with 5-FC to the one obtained in SD without 5-FC in each condition. Statistical significance is as follows: ****, adj. *p*-value < 0.0001; ns, not significant. C) Competition assay. Individual cultures of the deletion mutants were prepared in SD without selective pressure, then pooled in equal volumes, serially diluted and plated on large Petri dishes containing SD with 1.56 µg/mL 5-FC. For each counted colony, the genotype (*fcy1Δ* or *fur1Δ*) was confirmed by PCR. For each background, the relationship between CFU count of either mutant and the inoculum concentration was assessed by a Chi-square test of independence, performed on the cumulative sum of counts obtained from two independent experiments. D) Inhibition assay. For each background, a culture of the *fcy1Δ* mutant was inoculated in SD 1.7% agar with or without 25 µg/mL 5-FC and poured on top of thin 2% agar plates containing the matching medium. A culture of the *fur1Δ* mutant was then spotted on top. Uncropped pictures are provided in S8 Fig.

## Discussion

### Efflux-mediated resistance to 5-FC in *rho⁻* mutants

We generated and analyzed the fitness of hundreds of independent 5-FC resistant mutants to identify the most frequent molecular signatures of resistance to this antifungal. Challenging a common assumption of the usual paths to 5-FC resistance, we show that around a third of mutants gained resistance through a previously unknown mechanism likely involving drug efflux rather than inactivation of key metabolic enzymes. This path to resistance, often referred to as a pleiotropic drug response, is well characterized for azole resistance [42–44]. Mutants which have lost mitochondrial function are no longer able to respire and display a specific *petite* morphotype on glucose-containing medium. Resistance in this case therefore comes with a tremendous tradeoff. In turn, they display enhanced efflux activity, which confers generalized resistance. To date, this resistance mechanism has not been described for 5-FC.

Here, we confirm that 5-FC treatment at least selects for *rho⁻* mutants, which are cross-resistant to 5-FC and fluconazole (Fig 3A). However, we find no significant correlation between rhodamine accumulation, indicative of increased efflux capacity through ABC transporters, and fitness in 5-FC (Fig 3D). This corroborates previous observations that 5-FC treatment, while selecting for *rho⁰* mutants, does not affect the expression of transcription factors such as Pdr1 or the related expression of ABC transporters [22]. On the other hand, we do observe a significant correlation between rhodamine accumulation and fitness in fluconazole for NC-02 mutants (Fig 3C). Since azole resistance has also been associated with MFS transporters [45], one possible explanation for the discrepancy would be that a mechanism overlapping both regulatory pathways is sufficient to confer resistance to both classes of antifungals. One study supporting this hypothesis reported that a substitution in the gene *MRR1* was sufficient to confer cross-resistance to 5-FC and azoles in clinical isolates of *C. lusitaniae* [25]. This gene encodes a transcriptional regulator, which when mutated upregulates the expression of the multidrug transporter encoding gene *MFS7*. Another study reported reduced susceptibility to azoles in 5-FC resistant isolates of several pathogenic fungi, including *C. albicans* [46].

Although future work is needed to uncover the exact mechanisms governing 5-FC resistance through efflux, it is plausible that pathogenic fungi could evolve along this path, in addition to the more commonly described resistance paths in the canonical pathway. Losing mitochondrial function would be unfavored in environments that require respiration, given the strong fitness cost in drug-free conditions. However prolonged treatments could render that evolutionary path more accessible, as suggested by reports of such selection in a clinical isolate of *N. glabrata* in response to fluconazole [47].

### *FUR1*-inactivating mutations confer 5-FC resistance in *rho⁺* mutants

Using WGS, we found that most *rho⁺* mutants gained resistance through *FUR1* inactivation. Interestingly, a large diversity of mutations was detected in this gene. In contrast, the MARDy database only lists two *FUR1* mutations conferring resistance to 5-FC at the time of writing

(version 1.1DB:1.3WS, http://mardy.dide.ic.ac.uk/). Despite the low number of reported mutations (including the ones reported in other studies but not listed in MARDy [11]), our experiment still recapitulates some of them. Most importantly, they correspond to mutations of clinical relevance, such as F211I and G190D found in clinical isolates of *C. auris* and *N. glabrata*, respectively [48,49]. Our work shows that the available paths through mutations in *FUR1* are far more diverse than the literature lets appear. Most mutations in *FUR1* appear to be loss-of-function mutations. We observed several indels (deletions and insertions of up to 49 and 39 bp, respectively) and five mutations affecting the translation start site of Fur1 (M1V, M1I, M1L, M1K, M1T) (S2 Data). Many more are predicted to impact the stability of the protein. This would make resistance through mutations in *FUR1* very likely since many of the possible amino acid changes are predicted to impact the stability of Fur1 and/or conserved regions (S4 Fig), in addition to all possible stop codons and other possible frameshifting indels. Not only the mutations picked up in our experiments capture both of these phenomena (S4 Fig), some also appear to likely affect proper assembly of the Fur1 tetramer (Fig 5C).

It is also possible that the drug concentration used to select for resistant mutants was high enough that a partially functional Fur1 would not confer resistance. It would be interesting to study the relationship between the extent of Fur1 inactivation and inhibitory concentrations to formally test this hypothesis. In any case, our results strongly suggest that the number of resistance-conferring mutations in fungal pathogens is profoundly underestimated. They do confirm however that resistance can emerge rapidly in response to 5-FC monotherapy [50]. That being said, rapid emergence of resistance does not seem to be specific to 5-FC, whether it is observed through experimental evolution or in the clinic. For example, cases have been reported where resistance through nonsense mutations in *FUR1* was rapidly acquired upon combination therapy with caspofungin [51]. Cross-resistance to fluconazole (and potentially other azoles) has also been reported to evolve rapidly in clinical settings [46]. The same can be said of nystatin resistance in experimental evolution [52]. Additionally, the fact that resistance to different antifungals is connected through shared resistance mutations despite their apparent lack of similarity in their mode of action is worrisome and stresses the need for more studies on cross-resistance.

## Alternative paths to resistance in *rho*⁺ mutants

Our experiment led to the identification of three other genes potentially conferring resistance to 5-FC when mutated: *URA6*, *GFA1* and *ARG2* (Fig 4C).

*URA6* encodes the enzyme which catalyzes the step right after the reaction performed by Fur1 (Fig 1), therefore it is highly likely that the detected mutations inactivate *URA6* and are sufficient to confer 5-FC and 5-FU resistance. A temperature-sensitive mutant of this enzyme in a laboratory strain is known to be resistant to 5-FU, confirming that loss of function is responsible for resistance [31]. *GFA1* encodes an important enzyme involved in the biosynthesis of chitin. Some studies have shown fungal cells can display increased chitin levels in response to fluconazole or echinocandins [53–55], however it appears to be one of many consequences of a generalized stress response. Both *URA6* and *GFA1* are essential in auxotrophic laboratory strains [31,56]. This shows that epistasis can play an important role in the evolution of resistance, and that the use of prototrophic strains in experimental evolution experiments will likely lead to different outcomes.

Finally, it has previously been shown that deletion of *ARG2* can confer resistance to 5-FU, through crosstalks between the ornithine biosynthesis pathway and the pyrimidine biosynthesis pathway [57]. The hypothesized mechanism is that blocking ornithine formation hinders the consumption of carbamoyl-phosphate, both being required for the synthesis of arginine.

As a result, carbamoyl-phosphate is used as an early precursor of UTP, which ends up being produced in larger quantities than 5-FUTP [57]. This would corroborate a previous observation that five other genes involved in arginine metabolism are linked with 5-FC resistance [23].

Our results highlight the need for studies using prototrophic strains, instead of the popular laboratory strains, as they allow the identification of rare alleles conferring resistance.

### *FUR1* is the main target to confer 5-FC resistance

Out of all genes involved in the metabolism of 5-FC, we unexpectedly detected mutations only in *FUR1* and *URA6* (Fig 4C). Resistance to 5-FC in clinically relevant fungi typically results from mutations in *FCY2*, *FCY1* and/or *FUR1* [17,58,59]. Since resistance often requires loss of function of those genes, redundant entry routes in *S. cerevisiae* can explain the absence of resistance mutations in genes involved in the import of 5-FC, namely *FCY2*, *FCY21*, *FCY22* and *FUR4* [13] as their loss of function would be masked by other, intact genes. The case of *FCY1* was more difficult to interpret as there is no known redundancy for its function. We hypothesized that depending on the context, *FUR1* could be a more likely target than *FCY1* due to cross-feeding. In a mix of *FCY1*+ and *fcy1*- cells, 5-FU would be produced and secreted, leading to the death of *FCY1*+ cells, as well as loss-of-function mutants. The benefits of losing the cytosine deaminase function would therefore be highly specific to the ecological context.

Accordingly, we performed a growth-based experiment in which the medium was previously conditioned by the growth of either a *fcy1*Δ mutant or a *fur1*Δ mutant (Fig 6B). A medium conditioned by the latter contains 5-FC, as well as 5-FU, as a result of deamination of 5-FC by Fcy1. 5-FU being toxic for any cell that encodes a functional Fur1, it imparts selective pressure upon *FUR1* mutants. We further confirm this hypothesis by showing that plating a mix of *fcy1*Δ and *fur1*Δ mutants at high cell density prevents the growth of *fcy1*Δ cells (Fig 6C). Finally, spotting the *fur1*Δ mutant on top of a *fcy1*Δ lawn leads to the apparition of an inhibition zone, delimiting the range of diffusion of 5-FU (Fig 6D and S8 Fig).

Compound sharing during the selection step dilutes the populations of cells in which mutations impact any step upstream of the conversion of 5-FU. This is of the utmost ecological importance, since the ecological context will define which mutations are more likely to lead to resistance [60].

### No strong background-dependent effect for path to resistance but many background-dependent effects on fitness

Even though we used two distinct environmental isolates as parental strains, we observe strong parallel evolution of resistance. Notably, the main outcome of our evolution experiment did not seem to be affected by the presence of active transposable elements in only one of the two parental strains (LL13-040). Indeed, for both backgrounds, most evolved strains gained resistance through loss-of-function mutations in *FUR1*. However, differences in fitness reflect the different behaviors of each background before and after evolution. For example, even though both parental strains grow at comparable rates in all tested conditions (S1 Fig), NC-02 displays more rhodamine accumulation than LL13-040 (Fig 3B). After evolution however, *rho*- strains of both backgrounds display strikingly similar distributions in rhodamine accumulation (Fig 3B). One background, NC-02, therefore evolved to gain much more efflux capacity than the other. This observation is corroborated by the significant correlation with fitness in fluconazole, which is again markedly higher for NC-02 *rho*- strains compared to LL13-040 counterparts (Fig 3C). Similarly, the same mutants show a lower fitness tradeoff in drug-free media (Figs 2 and 3A). Together, our results could indicate that NC-02 would potentially evolve

along a broader range of paths to resistance. Interestingly, this hypothesis is also supported by the mutations detected in the sequenced genomes. For example, more indels were detected in *FUR1* for NC-02 strains (Fig 5A). Two genes potentially sufficient to confer resistance, *GFA1* and *ARG2*, were found to be mutated only in NC-02 strains (Fig 4C). On the other hand, a single mutation in Fur1 (R110G) was found to be present in 39 different LL13-040 strains (S3 Fig). And despite selection seemingly acting more strongly in one background than the other, we found an identical number of unique mutations in *FUR1* for each background (52 background-specific mutations for each background and 14 shared mutations, S2 Data and S5 Fig).

### Experimental design and evolutionary relevance

Our experimental design does not represent a real-world clinical situation in which 5-FC would be used for treatment. However, we revealed a very interesting community interaction that can affect the route of evolution. Upon treatment of an infection, rapid establishment of resistance would arguably be impacted by two main factors: the local concentration of antifungal and the local population size. The first factor determines for instance the efficiency of selection and the second the size of the mutational supply and the efficiency of selection. Our results indicate that population size or density could also affect the route of evolution.

Drug concentration is rarely homogeneous, but instead exists as a spatiotemporal gradient linked to the inherent pharmacodynamic properties of the drug, such as imperfect drug penetration [61]. This can have a crucial effect on the establishment of resistance. For example, it can allow (or not) the expansion of a resistance-conferring mutation from a single cell to a whole subpopulation [62]. Drug concentration has also been shown to impact the nature of beneficial mutations and affect whether a population will develop resistance or tolerance [63]. Local population size is also heterogeneous. For example, a localized higher density of cells could be the result of a biofilm having developed on a catheter [8,64]. Our findings suggest that the benefits of losing either of two consecutive steps in a path, both of which confer resistance on their own, will be dictated by the local conditions and density. In conditions that allow cross-feeding and diffusion of small molecules, the loss of the initial step (Fcy1) would not be advantageous. However, for isolated cells or in conditions where the diffusion of nutrients is highly limited, it might be, hence the discrepancy in the resistance phenotype of the *fcy1*Δ mutant on a shared plate versus in an isolated well (S7 Fig). In comparison, the loss of the second step (Fur1) is advantageous, no matter if cells are isolated or in close contact (Fig 6C).

Ultimately, evolution experiments such as our study highlight relatively narrow but still very relevant features of evolution of resistance. And even with the complex challenge that is the accurate reproduction of clinically relevant environments, these studies still manage to capture resistance mutations found in clinical isolates [44].

## Materials and methods

### Strains, plasmids, primers and culture media

Strains used in this study are listed in S1 Data. We used two environmental strains of *S. cerevisiae*: LL13-040 [65] and NC-02 [66], both wild isolates collected from trees in North America. Unlike LL13-040, previous whole-genome analysis of NC-02 revealed no active transposable element [66], which could potentially change the types of resistance mutations the strains would have access to. For both strains, only haploids were used to be able to detect recessive mutations. First, the *HO* locus was replaced by a nourseothricin resistance marker in LL13-040 and a hygromycin resistance marker in NC-02. Strains underwent sporulation and dissection. Haploid selection and mating type were confirmed by multiplexed PCR. All subsequent

experiments were performed using the corresponding haploid strains of mating type MAT**a**. They are referred to thereafter as the "parental" strains in our experimental evolution. The two strains have 0.39% nucleotide divergence (averaged across chromosomes).

Strain construction was done by standard transformation from competent cells using plasmids and primers listed in S1 Data. The following culture media were used: YPD (1% yeast extract, 2% bio-tryptone, 2% glucose, with or without 2% agar), YPG agar (1% yeast extract, 2% bio-tryptone, 3% glycerol, 2% agar), SD (MSG) (0.174% yeast nitrogen base without amino acids, 2% glucose, 0.1% monosodium glutamate), SD ($NH_4$) (0.174% yeast nitrogen base without amino acids, 2% glucose, 0.5% ammonium sulfate), SC (SD with standard drop-out mix). Most biochemical products were acquired from Fisher Scientific or BioShop Canada. When indicated, the following compounds were added to the medium: 5-FC (Fisher Scientific), 5-FU (Fisher Scientific), cytosine (Fisher Scientific), methylcytosine (Fisher Scientific), micafungin (Toronto Research Chemicals), caspofungin (Cedarlane Labs), nystatin (BioShop Canada), fluconazole (Cedarlane Labs), nourseothricin (Millipore Sigma), hygromycin B (BioShop Canada), G418 (BioShop Canada).

## Experimental evolution

The evolution experiment was adapted from [67] to isolate a large number of independent 5-FC resistant mutants. Parental strains were streaked from glycerol stocks onto YPD agar medium, then incubated three days at 30˚C. 10 mL YPD was inoculated with a single colony, then incubated for 24 hours at 30˚C with shaking. The corresponding number of mitotic generations was calculated from measurements of cell concentrations before and after incubation using cell counts estimated by flow cytometry for four biological replicates, with a Guava easy-Cyte HT cytometer (Cytek, blue laser): LL13-040, 5.3 ± 0.5 generations; NC-02, 5.4 ± 0.5 generations. Each 24 h preculture was used to inoculate a 96-deep-well plate with 1 mL synthetic minimal medium (SD) supplemented with 1.56 μg/mL 5-FC, at a final concentration of 0.1 $OD_{600}$. Border wells were filled with 1 mL sterile water to prevent evaporation. Plates were sealed with porous adhesive film and incubated for 72 hours at 30˚C with shaking. Finally, cultures were spotted on SD agar plates with or without 6.25 μg/mL 5-FC to select 5-FC resistant mutants. All mutants were streaked on YPD agar medium to isolate single colonies, only one of which per mutant was selected for further experiments.

*rho*⁻ (*petite*) mutants were identified by their morphotype. Once enough mutants were generated, a 96-well plate with 1 mL YPD was inoculated and incubated overnight at 30˚C. The plate was then used to prepare a glycerol stock, as well as spotting on YPG agar medium to confirm *rho* status. Overall, the evolution experiment was iterated eight times, for a total of 49 precultures (26 for LL13-040 and 23 for NC-02), generating a total of 682 5-FC resistant mutants (296 LL13-040 mutants and 386 NC-02 mutants). Two more iterations without 5-FC selection were performed to evaluate whether acclimation to minimal medium could be enough to acquire resistance. It was not, since neither generated any 5-FC resistant mutant.

Media conditions for the evolution experiment were initially determined based on preliminary growth assays using the constructed haploid LL13-040 strain. Specifically, we noted that the use of minimal medium greatly increases 5-FC sensitivity, the absence of uracil being partly responsible for this effect (S9A Fig). The final concentration of 5-FC used in the evolution experiment was chosen based on dose-response curves in SD medium obtained from biological triplicates of both strains. In the conditions of this experiment (see section below), a concentration of 1.56 μg/mL 5-FC corresponds to inhibition coefficients of 93.5 ± 0.8% and 91.8 ± 0.3% for LL13-040 and NC-02, respectively (S9B Fig). In an initial experimental evolution trial, this concentration yielded 15 LL13-040 spontaneous mutants, compared to only 3

when using 0.78 μg/mL 5-FC. A 10% cutoff (at least six mutants out of 60 wells inoculated on each plate) ultimately prompted us to use 1.56 μg/mL 5-FC for all iterations of the evolution experiment.

## Automated growth measurements on solid medium

The growth of resistant strains, as well as parental strains and deletion mutants for *FCY1* and *FCY2* was assayed with a BM3-SC robot (S&P Robotics Inc.). First, plates of mutants were printed on YPD OmniTrays. Then, mutants were rearrayed depending on their *rho* status. Finally, the plates were replicated onto solid medium, either rich medium, or minimal medium with or without antifungal. The following sections describe in more detail the protocol followed depending on the *rho* status.

*rho*⁺ **mutants.** A liquid culture of strain s_009 was used to print a border on a YPD 384-array. 300 *rho*⁺ mutants (including controls) were then rearrayed onto the same 384-array from five 96-array sources. The resulting 384-array was expanded into a 1536-array to have four replicates for each mutant. The 1536-array was replicated six times to obtain the source plates for the fitness measurements. Source 1 was replicated onto five destination plates: YPD, SD, SD + 25 μg/mL 5-FC, SD + 1.56 μg/mL 5-FU, SD + 6.25 μg/mL 5-FU, which were incubated for 22 h at 30°C. Sources 2–6 were each replicated onto three destination plates for a total of 15 conditions: YPD, SD, SD + 25 μg/mL 5-FC, SD + 16/32/64 μg/mL fluconazole, SD + 4/8/16 μg/mL nystatin, SD + 0.25/1/2 μg/mL caspofungin and 0.0625/0.25/0.5 μg/mL micafungin. Plates were incubated for 90 h at 37°C. During both incubations, pictures of each plate were taken every 2 h in a spImager custom robotic platform (S&P Robotics Inc.).

*rho*⁻ **mutants.** Similarly, two 96-arrays were prepared, one with only LL13-040 mutants, the other with only NC-02 mutants. For both, a liquid culture of the corresponding *fcy1*Δ mutant was used to print the border. 118 *rho*⁻ mutants as well as controls were rearrayed from either three (NC-02) or four (LL13-040) 96-array sources. The two resulting 96-arrays were expanded into two 384-arrays to have four replicates for each mutant. For each background, the 384-array was replicated five times to obtain the source plates for the fitness measurements. Similar growth media conditions were tested. Additionally, both plates were replicated onto YPG arrays to confirm *rho* status. Following this quality control step, four mutants were detected as being initially misannotated as *rho*⁻ (three LL13-040 mutants and one NC-02 mutant) and were therefore excluded from all analyses (S10 Fig).

**Colony size analyses.** Pictures were cropped, then converted into inverted gray levels using scikit-image [68]. Colony sizes were quantified using pyphe-quantify in batch mode [69]. The area parameter was used to generate growth curves and calculate the corresponding area under the curve (AUC) using the composite trapezoidal rule. For each mutant, the relative fitness was calculated as the ratio of the mean AUC (absolute fitness) divided by the corresponding mean AUC obtained for the WT parental strain.

## Cytometry

**Rhodamine accumulation assay.** In order to evaluate the efflux capacity of *rho*⁻ mutants, we used rhodamine 6G, a fluorescent dye known to be a substrate of ABC transporters. Measurement of intracellular accumulation of rhodamine was adapted from [70]. The two 96-arrays of *rho*⁻ mutants for LL13-040 and NC-02 (obtained as described above) were inoculated in 96-deep-well plates containing 1 mL YPD and incubated overnight with shaking at 30°C. For both plates, two controls were added: the parental strain and a strain with deletions for all ABC transporters (*PDR5 SNQ2 YBT1 YCF1 YOR1*, s_012 in S1 Data) [71], therefore incapable of rhodamine efflux. In the morning, both plates were subcultured at 0.15 $OD_{600}$,

then incubated at 30˚C with shaking until they reached 0.6 $OD_{600}$. For this step, one of the mutants was subcultured twice to later have a control without rhodamine. 200 uL was transferred to a sterile V-shaped plate. Rhodamine 6G (Millipore Sigma) was added to a final concentration of 10 μg/mL except for 1 well. Plates were sealed with porous adhesive film and incubated 30 min at 30˚C with shaking. Cultures were centrifuged 5 min at 230 g and pellets were washed with sterile PBS. Cultures were centrifuged again and pellets were resuspended in 200 μL sterile PBS with 0.2% glucose. Cultures were incubated an extra 30 min at 30˚C to activate energy-dependent efflux, then diluted 1:10 prior to fluorescence measurements by flow cytometry. Fluorescence in the green and orange channels using blue and green lasers, respectively, was acquired for approximately 2,000 events per sample with a Guava easyCyte HT cytometer (Cytek). Fluorescence values in both channels were normalized with the cell size using the FSC value. Thresholds were set for both normalized fluorescence values to maximize events with fluorescence below the thresholds for the negative control (sample not treated with rhodamine), as well as maximize events with fluorescence above the thresholds for the positive control (s_012 in S1 Data). The percentage of cells with fluorescence above both thresholds is considered to be inversely correlated with the efflux capacity.

**Fluorescence reporter assay.** A GFP reporter was used to evaluate the level of expression of *FCY1* in one of the parental strains to assess *FCY1* functionality. Precultures of LL13-040 with or without *FCY1-mEGFP* (s_002 and s_003 in S1 Data) were prepared from single colonies in 5 mL YPD medium and incubated overnight at 30˚C with shaking. Precultures were diluted at 1.5 $OD_{600}$ in sterile water. 24-well plates containing either SC, SC -ura, SD (MSG) or SD ($NH_4$) with 0, 0.78 or 1.56 μg/mL 5-FC were inoculated at 0.15 $OD_{600}$, sealed with porous adhesive film and incubated for 4 h at 30˚C with shaking. Cultures were diluted at 0.05 $OD_{600}$. Fluorescence in the green channel using a blue laser was acquired for 5,000 events. Events with a SSC value (proxy for cell granularity) below 300 or a FSC value (proxy for cell size) below 2,000 were filtered out. The fluorescence values were normalized by the FSC value.

## DNA extraction and sequencing

In order to identify which mutation(s) were conferring 5-FC resistance in $rho^+$ mutants, we performed whole-genome sequencing. The mutants were cultured overnight at 30˚C in 24-well plates with 2 mL YPD, sealed with porous adhesive film. Cells were pelleted from a total of approximately 20 OD units per sample. DNA was extracted using the MasterPure Yeast DNA Purification kit (Epicentre) following the kit's protocol, except for the following adjustments. Isopropanol precipitation was performed for 1 h at room temperature. Pellets were dried at 55˚C for 25 min, after which they were resuspended in 50 μL 0.2 ng/μL RNase. Samples were incubated an additional 5 min at 55˚C, then immediately purified using SPRI beads (1:1 ratio, Axygen). Attempts to extract genomic DNA from $rho^-$ mutants proved unsuccessful with this method.

Genomic DNA was stored at -20˚C, then quantified in technical triplicates using the Accu-Clear Ultra High Sensitivity dsDNA Quantitation kit. A total of 282 samples (276 out of 294 rearrayed $rho^+$ mutants, as well as the parental strain and the *fcy1*Δ and *fcy2*Δ mutants for each background, s_004, s_005, s_009, s_010) were diluted to 10 ng/μL for a total of 40 ng and arranged on three 96-well plates. For eight samples, whose initial concentration was too low, a volume of 15 μL was pipetted and dried in a SpeedVac, then resuspended in the appropriate volume to the desired final concentration of 10 ng/uL. Sequencing libraries were prepared using the Riptide High Throughput Rapid DNA Library Prep kit (iGenomx), according to the kit's protocol. The three 96-well plates were each pooled separately. For the amplification step, we modified the protocol to use barcoded primers for each pool instead of a single universal

PCR primer. The three pools were SPRI-purified in two steps with protocol option 3 (recommended for PE150 sequencing). The pools were checked on agarose gel, quantified with the AccuClear Ultra High Sensitivity dsDNA Quantitation kit and analyzed on BioAnalyzer (Agilent). Finally, they were sequenced in paired-end 150 bp on an Illumina NovaSeq 6000 S4 system at the Centre d'expertise et de services Génome Québec.

*FCY1* and *FUR1* were amplified from the same genomic DNA using primers listed in S1 Data, then sequenced by Sanger amplicon sequencing. For *FUR1* amplicons, all samples for which enough material could be recovered (after preparing WGS libraries) were processed. The sequencing reactions were performed at the Centre Hospitalier de l'Université Laval sequencing platform (Université Laval, Québec).

## Validations and complementations

Once we identified *FUR1* as being the hotspot for mutations in *rho*+ mutants, we confirmed the detected mutations were causal by using two types of growth assays: *validation* refers to a growth assay in isolated liquid medium where we compared the growth of mutants to that of the *fur1*Δ mutant, whereas *complementation* refers to measuring the growth of mutants upon expression of the WT allele of *FUR1* carried by a plasmid.

**Validations.** 35 *rho*+ mutants (LL13-040; n = 17, NC-02, n = 18), the deletion mutants for *FCY1*, *FCY2* and *FUR1* (s_004, s_005, s_006, s_009, s_010, s_011 in S1 Data) and both parental strains were precultured in duplicates in a single 96-deep-well plate with 1 mL YPD overnight at 30˚C with shaking. Precultures were diluted at 1 $OD_{600}$ in sterile water and inoculated at a final concentration of 0.1 $OD_{600}$ in two sterile Greiner 96-well plates with SD or SD + 100 μg/mL 5-FC. $OD_{600}$ measurements were performed at 30˚C every 15 min until a plateau was reached in a Tecan Infinite M Nano (Tecan Life Sciences).

**Complementation.** 32 out of the 35 validation mutants (16 from each background) and the *fur1*Δ mutants were transformed with pMoBY or pMoBY-*FUR1* [72] using a modified yeast transformation method [73]. Transformants were selected with combinations of nourseothricin or hygromycin B and G418. Precultures were prepared in a single 96-deep-well plate with 1 mL YPD + G418, inoculated from a single colony for each strain and incubated overnight at 30˚C with shaking. Precultures were diluted at 1 $OD_{600}$ in sterile water and inoculated at a final concentration of 0.1 $OD_{600}$ in a sterile Greiner 96-well plate with SD + 100 μg/mL 5-FC. $OD_{600}$ measurements were performed at 30˚C every 15 min until a plateau was reached in a BioTek Epoch 2 Microplate Spectrophotometer (Agilent).

## Growth assays in isolated liquid cultures

Three types of growth assays in isolated liquid cultures were carried out. 1- 'Small-scale' assays (precultures in 5 mL tubes and growth measurements in 96-well plates) were used a) to evaluate 5-FC sensitivity in different media (minimal and rich) and b) to measure the growth of parental strains in minimal medium with different sources of nitrogen. 2- A single 'larger-scale' growth assay (precultures in 96-well plates and growth measurements in 384-well plate) was used to measure the growth of a subset of *rho*+ mutants, as well as the *fcy1*Δ mutant, in liquid medium, in order to compare with the growth measured on solid medium. 3- A dose-response curve assay was used to measure the inhibition coefficient corresponding to the 5-FC concentration used in the evolution experiment.

**'Small-scale' growth assay.** For the 'small-scale' growth assays, strains were streaked on YPD agar medium supplemented with the appropriate antibiotics. Precultures were prepared from three isolated colonies in 5 mL YPD medium with appropriate selection and incubated overnight at 30˚C with shaking. Precultures were diluted at 1 $OD_{600}$ in sterile water. Sterile

Greiner 96-well plates were prepared with the indicated culture medium and compounds and cultures were inoculated at a final concentration of 0.1 $OD_{600}$. $OD_{600}$ measurements were performed at 30˚C every 15 min until a plateau was reached in a Tecan Infinite M Nano (Tecan Life Sciences).

**'Larger-scale' growth assay.**    For the 'larger-scale' growth assay, precultures were prepared in a 96-deep-well plate with 1 mL YPD and incubated overnight at 30˚C with shaking. Precultures were diluted at 1 $OD_{600}$ in sterile water. A sterile 384-well plate containing four different conditions was used: SD, SD + 25 μg/mL 5-FC, SD + 100 μg/mL 5-FC and YNB + 2% glucose + 250 μg/mL cytosine. The lid was conditioned by incubating 3 min with 5 mL 0.05% Triton X-100 / 20% ethanol and dried under sterile conditions. Cultures were inoculated in a single replicate at a final concentration of 0.1 $OD_{600}$ in a final volume of 80 μL per well. $OD_{600}$ measurements were performed at 30˚C every 15 min in a Tecan Spark (Tecan Life Sciences) with an active Tecool temperature control module, until curves showed signs of evaporation (13 h).

**Dose-response curve assay.**    A single 96-well plate was prepared as described for the 'small-scale' growth assays, with cultures of LL13-040 and NC-02 in biological triplicates. 5-FC was serially diluted 1:2 seven times starting from a final concentration of 3.125 μg/mL. The maximum growth rate was transformed into the inhibition coefficient, with an inhibition coefficient of 0 corresponding to the maximum growth rate measured in the absence of 5-FC.

## Assessment of cross-feeding-induced toxicity

Three types of growth-based assays were carried out to investigate the selection of $fur1^-$ mutants over $fcy1^-$ mutants. A liquid medium conditioning assay was used to evaluate if a toxic compound was secreted by the $fur1\Delta$ mutant. A competition assay was used to assess the importance of cell density in liquid medium. Finally, an inhibition assay was performed to confirm the phenotype on solid medium.

**Medium conditioning assay.**    Precultures of LL13-040 and NC-02 with or without deletion of *FCY1* or *FUR1* (s_002, s_004, s_006, s_008, s_009, s_011) were prepared from three isolated colonies in 5 mL YPD medium and incubated overnight at 30˚C with shaking. Precultures were diluted at 1 $OD_{600}$ in sterile water. Subcultures of 5 mL SD medium with or without 1.56 μg/mL 5-FC in 50 mL Falcon tubes were inoculated at a final concentration of 0.1 $OD_{600}$. After a 5h incubation at 30˚C with shaking, cultures had reached 0.5–0.6 $OD_{600}$. Cultures were centrifuged 5 min at 500 g and the supernatants were filtered, then transferred to a sterile Greiner 96-well plate. The media "conditioned" by the growth of the $fcy1\Delta$ mutant were inoculated with either the *FUR1* null mutant or the WT. The media conditioned by the growth of the *FUR1* null mutant were inoculated with either the $fcy1\Delta$ mutant or the WT. Cultures were inoculated at a final concentration of 0.1 $OD_{600}$. Growth curves were acquired as described above.

**Competition assay.**    Precultures of LL13-040 and NC-02 with or without deletion of *FCY1* or *FUR1* (s_002, s_004, s_006, s_008, s_009, s_011) were prepared as described above. Precultures were diluted at 1 $OD_{600}$ in sterile water. Subcultures of 5 mL SD medium without 5-FC were inoculated at a final concentration of 0.1 $OD_{600}$. After a 5h incubation at 30˚C with shaking, cultures had reached 0.6 $OD_{600}$. For each background, cultures of both deletion mutants were pooled in equal volumes and serially diluted in sterile water. 0.5 mL of each dilution ($10^{-3}$, $10^{-4}$ and $10^{-5}$ OD) was plated on large Petri dishes containing SD with 1.56 μg/mL 5-FC. From each plate, 16 colonies were streaked on the same medium to make sure they were 5-FC resistant. Additionally, the genotype ($fcy1\Delta$ or $fur1\Delta$) was identified by multiplexed colony-PCR using one common primer annealing in either marker (*NAT* or *HPH*) and two gene-specific primers annealing upstream the deletion site. The reported CFU counts correspond to

the mean of 2 replicate experiments, but the statistical test (Chi-square of independence) was performed on the cumulative sum of counts.

**Inhibition assay.** Precultures of LL13-040 and NC-02 with or without deletion of *FCY1* or *FUR1* (s_002, s_004, s_006, s_008, s_009, s_011) were prepared from three isolated colonies in 5 mL YPD medium and incubated overnight at 30˚C with shaking.

## Computational analyses

**WGS data.** FASTQ reads were demultiplexed (and adapter-trimmed) using DemuxFastqs from the fgbio set of tools. A custom snakemake pipeline was used to process the FASTQ files [74]. Reads were aligned using bwa-mem2 on the reference S288C genome [75]. Aligned reads were sorted and indexed using samtools [76]. Reads that did not map uniquely (SAM FLAG 256) were excluded. PCR duplicates were removed using the picard function MarkDuplicates. Reads around indels were realigned using samtools calmd. The pipeline was run on the IBIS servers.

**Variant calling.** Variants were called with two methods, bcftools/samtools and GATK.

**SNP calling with samtools/bcftools ("samtools").** SNP calling was conducted with bcftools v1.9. SNP pileup was generated with bcftools mpileup command with options -C50, -min-MQ 4, -min-BQ 13 and prior removal of reads that were unmapped, not in primary alignment, failing quality checks or were PCR/optical duplicates. The command was run separately for samples from the two backgrounds (LL13-040 and NC-02). Haploid SNP calling was conducted with bcftools call command with option -mv.

**SNP calling with gatk ("gatk").** In the second variant calling method, bam files were edited using picard v2.18 (http://broadinstitute.github.io/picard/), SNP and indel calling was conducted with GATK v4.2.6.1 [77]. Prior to variant calling, an RG (read group) tag was added to individual bam files. GVCF files were generated with GATK HaplotypeCaller, with options -ploidy 1 -ERC GVCF and—min-base-quality-score 20. GVCF files of samples coming from the same background (LL13-040 or NC-02) were combined with GATK CombineGVCFs, genotyped with GATK GenotypeGVCFs, and processed separately. SNPs with the following criteria were filtered out: variant quality score (QUAL) < 30, QUAL by depth (QD) < 2, mapping quality (MQ) < 40, Fisher's exact tests of strand bias (FS) > 60, symmetric odds ratio test of strand bias (SOR) > 4, mapping quality rank sum test (MQRankSum) < -12.5, and rank sum test for site position within reads (ReadPosRankSum) < -8. Indels were filtered out if they met the following criteria: QD < 2, QUAL < 30, FS > 200, or ReadPosRankSum < -20.

Further filtering was conducted for all variant sets with bcftools v1.9 and python v3.10 scripts. We applied a set of filtering criteria both to samples, and variants. First, we removed samples (marked as missing data), which met the following criteria: read depth < 4, allelic depth (AD) for the second most common allele > 4, ratio of the AD of the second to the first most common allele > 0.2. Then, variants meeting the following criteria were removed: mean read depth across all samples < 10, total read depth > 20000, variant quality (QUAL) < 20, mapping quality (MQ) < 40, high frequency (allele frequency (AF) > 0.99 or allele count (AC) = 0). Finally, any variants shared with the parental strain were removed as well. Variants were ultimately annotated with SnpEff v5.0 [78] using *S. cerevisiae* genome vR64-3-1.

**CNVs.** Read coverage of WGS data was analyzed to look for regions with copy number variations (larger than 5,000 bp and smaller than half of the chromosome length). Read count per position for each sample was output using samtools depth. Mean read count was calculated in non-overlapping windows of 5,000 bp and corrected for mapping bias related to GC content. Namely, the read count in each window ("W") was multiplied by a ratio of median read count over all windows divided by a median read count over all windows with the same GC as

"W". End chromosome bias (increasing read count towards chromosome ends) was corrected by fitting a curve for the windows closest to one chromosome end (each half of the chromosome), using the LOWESS smoothing method, in python v3 statsmodels package, with a default of 2/3 of data points to estimate each y value. Windows were then divided by the fitted line to obtain the normalized read count.

**Predictions of impact of mutations.** Mutations were submitted to the online tool mutfunc to predict their impact, notably on stability and/or conservation in a homology model [79].

## Supporting information

**S1 Fig. Growth of individual mutants on solid medium.** Growth corresponds to the mean area under the curve (AUC, calculated on 22 h) from four replicate colonies, for individual strains arrayed on solid media: (A) YPD, SD, SD + 25 µg/mL 5-FC and SD + 6.25 µg/mL 5-FU, incubated at 30˚C and (B) YPD, SD, SD + 25 µg/mL 5-FC, SD + 0.5 µg/mL micafungin, SD + 2 µg/mL caspofungin, SD + 16 µg/mL nystatin and SD + 64 µg/mL fluconazole, incubated at 37˚C. Data correspond to Figs 2 and 3A before normalization with the WT, here represented as split violin plots. For both growth assays (corresponding to panels A and B), all $rho^+$ strains were gathered on a single plate, with WT controls for the two backgrounds (LL13-040 and NC-02). The mean AUC for the WT controls in each condition is indicated by a gray diamond. Similarly, the mean AUC for the WT controls present on plates with $rho^-$ strains (in this case, one plate for LL13-040 $rho^-$ strains and another for NC-02 $rho^-$ strains) is indicated by a red diamond.
(TIF)

**S2 Fig. Depth per position.** Normalized coverage calculated for 5 kb windows. Each track corresponds to a sequenced genome. Red signal indicates a normalized coverage above 4X. LL13-040 genomes are displayed first (B10, B11. . .), then NC-02 genomes (B14, B15. . .).
(TIF)

**S3 Fig. Diversity of Fur1 mutations per preculture.** Each marker (dot or triangle) represents a single preculture, from which mutants were selected (# genomes). A gray dashed line indicates if as many mutations have been identified in Fur1 as the number of genomes that carried them. For one outlier (only three mutations found in 41 strains which arose from the same preculture), a pie chart details the corresponding mutations and the number of strains that carried them.
(TIF)

**S4 Fig. Predicted impact of Fur1 mutations.** Effect on stability (A) and conservation (B) of all possible amino acid substitutions in Fur1 which are predicted to be impactful by mutfunc [79]. Substitutions predicted to be non-deleterious are not shown. The substitutions captured in our dataset are highlighted (bigger colored dots). (A) Side plots indicate the kernel densities for all data points (lightgray) and substitutions captured in our dataset (black) along the position in Fur1 (top) or the ΔΔG value (right). (B) SIFT scores are represented on an inverted y-axis and are analogous to a $p$-value. Scores < 0.05 (gray dashed line) indicate a predicted deleterious mutation, with a low value (top of the plot) indicating that the amino acid change is very likely to affect protein function based on sequence conservation. For substitutions captured in our dataset, the color indicates the conservation level of the residue at that position.
(TIF)

**S5 Fig. Location and type of detected mutations along the *FUR1* sequence.** The location of all detected mutations in the *FUR1* gene sequence (n = 118, length of the gene represented by a gray half-arrow) is indicated by a barplot (first track), where every bar represents a unique mutation, and their height represents the number of unique genomes in which the mutation was identified. Bars are color-coded to indicate in which background the mutation was identified. The second and third tracks contain boolean indicators for the method of detection and the type of mutation (black for true, gray for false). On the third track, the corresponding positions along the Fur1 protein sequence are indicated.
(TIF)

**S6 Fig. Fcy1 functionality in parental strains.** A) Density plots showing the fluorescence signal of *FCY1*-mEGFP (green), compared to the control without fluorescent reporter (gray) in the background LL13-040. Signal was acquired by cytometry for 5,000 events. A single threshold (gray dashed line) was used to indicate relative percentages of events for both strains in each condition. B) Growth was measured for both parental strains in YNB + 2% glucose containing different nitrogen sources.
(TIF)

**S7 Fig. Fcy1-mediated resistance in liquid and solid media.** A) Growth assay in liquid medium. Cultures were inoculated from single replicates in a 384-well plate containing SD (control), SD + 25 μg/mL 5-FC (5-FC) or YNB + 2% glucose + 250 μg/mL cytosine (cytosine). Relative growth corresponds to the area under the curve (AUC, calculated on 13 h) normalized by the WT. B) Relative growth measured in liquid medium (data from panel A) compared to the one measured on solid medium for the corresponding mutants (data from Fig 2) at equal concentrations of 5-FC.
(TIF)

**S8 Fig. 5-FU cross-feeding inhibition zones.** Uncropped pictures as shown in Fig 6D.
(TIF)

**S9 Fig. 5-FC dose response.** A) Growth assays for LL13-040 in different media supplemented or not with either 1.56 or 3.125 μg/mL 5-FC: synthetic complete medium with standard dropout mix (SC complete), SC without uracil (SC -ura), SD (MSG) with or without 0.2% aspartate and SD (NH$_4$) with or without 0.2% aspartate / 0.2% glutamate. The area under the curve (AUC) parameter was calculated from three biological replicates. B) 5-FC dose-response curves in SD (MSG) for LL13-040 and NC-02. The concentration is shown on a log2 scale. 0 μg/mL 5-FC was used to normalize growth values. For each strain, the mean of three biological replicates for each concentration was used to fit the Hill equation. The corresponding IC50 and Hill coefficients are indicated.
(TIF)

**S10 Fig. Growth measurements on YPG agar medium.** A, B) Pictures of arrays on YPG agar medium after 22 h incubation at 30°C for LL13-040 (A) and NC-02 (B). Each mutant is spotted in four replicates with the *fcy1Δ* mutant occupying the border as positive control. Gray squares highlight mutants initially misannotated as *rho*⁻. Pictures were cropped and converted into inverted gray levels for clarity and downstream analysis of colony size. C) Growth curves were obtained by automatic detection of colony size on transformed pictures taken every 2 h for 22 h at 30°C. Relative growth corresponds to the mean area under the curve (AUC) normalized by the WT. *rho*⁻ mutants included in all other figures are colored in red. Gray dots correspond either to the WT control (on the dotted line corresponding to a relative fitness of 1), or to the misannotated mutants mentioned above, which were therefore excluded from all analyses

except the rhodamine accumulation experiment.
(TIF)

**S1 Data. Strains, plasmids and primers used in this study.**
(XLSX)

**S2 Data. List of mutations detected in *FUR1*.** Columns: SNP_pos_aa, 1-based position in the protein sequence of Fur1; mutation; mutID_true, unique mutation identifier (convertible to mutfunc-friendly format); mutation_type, SNP or indel; POS, position in chrVIII; pos_nt, 0-based position in the *FUR1* gene; REF, reference residue; ALT, alternative residue; detected_by, indicates if the mutation was detected by samtools, gatk or Sanger (1 row per detection method); RA_well, unique strain identifier and location on the plate after rearray; background; fluc_assay, iteration of the evolution experiment; pre_culture, indicates if the mutant was generated from the same preculture; Genomix_plate_nb, indicates on which plate the genomic DNA was located for WGS; val_comp, indicates if the mutant was tested in the validation assay and/or the complementation assay.
(XLSX)

**S3 Data. Resistance-conferring mutations and their relative fitness at 30˚C.** Columns: resistance gene; mutation; medium (YPD, SD, SD + 5-FC or SD + 5-FU); # strains, number of strains in which the mutation was detected across both backgrounds; median relative fitness.
(XLSX)

**S4 Data. Illumina indexes and barcodes used for demultiplexing.** The master pool was demultiplexed using the i7 index at Genome Québec. The i5 index is provided but was not used for demultiplexing. Samples were demultiplexed from pools 1–3 using DemuxFastqs with read structure 8B12M+T 8M+T. Columns: pool; i7; i5; BioSample accession number; RA_well, unique strain identifier and location on the plate after rearray; sample barcode.
(XLSX)

## Acknowledgments

We thank Justin C. Fay for the kind gift of NC-02, Philippe Després for initial insights on pyrimidine cross-feeding, Mathieu Hénault for technical assistance, Isabelle Gagnon-Arsenault for critical reading of the manuscript and members of the Landry lab for feedback on the project.

## Author Contributions

**Conceptualization:** Romain Durand, Christian R. Landry.

**Data curation:** Romain Durand, Anna Fijarczyk.

**Formal analysis:** Romain Durand, Anna Fijarczyk.

**Funding acquisition:** Christian R. Landry.

**Investigation:** Romain Durand, Jordan Jalbert-Ross, Alexandre K. Dubé.

**Methodology:** Romain Durand, Anna Fijarczyk, Alexandre K. Dubé, Christian R. Landry.

**Project administration:** Christian R. Landry.

**Resources:** Christian R. Landry.

**Software:** Romain Durand, Anna Fijarczyk.

**Supervision:** Romain Durand, Alexandre K. Dubé, Christian R. Landry.

**Validation:** Romain Durand, Alexandre K. Dubé.

**Visualization:** Romain Durand, Anna Fijarczyk.

**Writing – original draft:** Romain Durand.

**Writing – review & editing:** Romain Durand, Anna Fijarczyk, Alexandre K. Dubé, Christian R. Landry.

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
