## [Decision Letter · Decision Letter 0]

31 May 2023

Dear Dr Durand,

Thank you very much for submitting your Research Article entitled 'Cross-feeding affects the target of resistance evolution to an antifungal drug' to PLOS Genetics.

The manuscript was fully evaluated at the editorial level and by independent peer reviewers. The reviewers appreciated the attention to an important problem, but raised some substantial concerns about the current manuscript. Based on the reviews, we will not be able to accept this version of the manuscript, but we would be willing to review a much-revised version. We cannot, of course, promise publication at that time.

You should specifically address concerns from Reviewers 1 and 3  about the use of fluctuation assays, or your definition of fluctuation assays. Reviewer 1 made some suggestions about how you could use fluctuation assay at lower cell densities for example. Reviewer 2 suggested that you directly test the role of Pdr3. There are several comments about the difficulty of following the data presented, and the supplementary material would benefit from reorganisation. 

If you decide to revise the manuscript for further consideration at PLOS Genetics, please aim to resubmit within the next 60 days, unless it will take extra time to address the concerns of the reviewers, in which case we would appreciate an expected resubmission date by email to plosgenetics@plos.org.

We are sorry that we cannot be more positive about your manuscript at this stage. Please do not hesitate to contact us if you have any concerns or questions.

Yours sincerely,

Geraldine Butler

Section Editor

PLOS Genetics

Geraldine Butler

Section Editor

PLOS Genetics

Reviewer's Responses to Questions

**Comments to the Authors:**

Reviewer #1: This study was designed to comprehensively elucidate the routes towards in vitro resistance to antifungal drug 5-fluorocytosine (5FC) in two different wild and prototrophic strains of S. cerevisiae. Selection for resistant mutants was done by starting with a preculture grown in YPD and followed by selection and enrichment for mutants in a 96-well format using SD medium supplemented with 1.56 μg/mL 5-FC. This procedure produced 682 5-FC-resistant mutants in the two strain backgrounds, about 1/3 of which were petite (non-respiring) and the rest had functional mitochondria. The authors showed that the petite 5-FC resistant mutants had also acquired resistance to fluconazole, consistent with extensive literature on the subject, and perhaps showed improved fitness in the echinocandin micafungin, although that was more difficult to conclude due to confounding growth effects. The vast majority of non-petite 5-FC resistant mutants had mutations in FUR1, which is in contrast to clinical mutations leading to 5-FC resistance in fungal pathogens, which encompass mutations in FUR1, FCY1, and FCY2 and also unexpected given the fact that the fcy1∆ mutant is highly resistant to 5-FC in their experimental system. The authors investigate this discrepancy and conclude that it’s due to the design of the in vitro evolution experiment, where any arising fur1 mutations would produce 5-fluorouracil that would be toxic to neighboring cells, including both wild type and fcy1 mutants, but not to the fur1 mutants themselves. They provide evidence for this conclusion by showing that in the medium preconditioned with a fur1 mutant but not with a wild type strain, fcy1 mutant no longer has a fitness advantage in the presence of 5FC. The major strengths of the study are the use of two different protrophic strains of S. cerevisiae and a clear description of the study design and its outcomes.

The major weakness, of which the authors appear to be aware, is that the in vitro evolution experiment produced a very narrow set of outcomes that do not correspond to those described in the clinical setting. The latter is not a weakness per se, as in vitro evolution experiments conducted in a model yeast are not expected to fully recapitulate the mutational signatures occurring in fungal pathogens occupying their host niches. However, in this particular case, any potential advantages of using S. cerevisiae for experimental evolution did not come to bear either, as the experimental design appeared to strongly select for mutations in FUR1 only. Given this outcome and the followup experiments, the two novel conclusions/hypotheses that can be drawn from this study are: 1. Petite strains with increased drug efflux may have increased fitness in 5-FC, which may be relevant clinically, and 2. Fungal cell density can affect the paths of evolution of antifungal drug resistance, e.g., via crossfeeding between fungal cells. Both hypotheses are potentially interesting for the field of antifungal drug resistance, but neither is explored very much. My suggestion for the authors, in order to make this study more convincing, thorough, and appropriate for PLoS Genetics, is to test the 2nd hypothesis by altering the setup of experimental evolution and showing that fcy1 mutations can now be obtained, e.g., at lower cell densities or when the strains carry a 2nd copy of FUR1. In this case, true fluctuation experiments (see comment below) can be useful to measure mutation rates and identify the genes responsible at different cell densities or when 2 copies of FUR1 are present. These experiments do not need to be done at large scale or necessitate additional costly WGS. Given the narrow range of resistant mutants, Sanger sequencing of candidate genes should be sufficient.

Weaknesses - minor:

1. The authors use the term “fluctuation assay” inappropriately. What they did was select for resistant mutations in the presence of 5-FC in liquid, whereas fluctuation experiments are done in drug-free liquid cultures, resulting in “fluctuating” numbers of spontaneous mutants that occur without selection and that are then detected as varying numbers of colonies on solid drug-containing medium.

2. How distantly related are the two S. cerevisiae strains?

3. In figure 3A, the asterisks indicate statistical significance; however, in some cases it is not clear what the differences between petite and non-petite strains are. For example, for LL13-040 in caspofungin, there are 4 asterisks, but it’s not clear whether rho+ or rho- strains have better fitness. Another way of showing the data is necessary to make that clear.

4. In figure 4C, I recommend replacing FCY1 and FCY2 with fcy1∆ and fcy2∆, as they are shown in 4D.

5. Sentence "All identified mutations in these three genes (n=7) are predicted to impact conserved residues with five out of seven also predicted to impact protein stability" (line 333) has not supporting evidence.

6. Sentence “…with all Sanger sequenced samples confirming what had been detected by WGS (Fig 5A)” (line 356) conflicts with what is actually shown in 5A: based on the upset plot, it appears that a few mutations detected by samtools and gatk were not detected by Sanger sequencing.

Reviewer #2: This study investigates the spectrum of spontaneous mutations conferring resistance to flucytosine (5-FC) in two different wild haploid isolates of S. cerevisiae. In both isolates, FUR1 deficiencies constitute the majority of the 5-FC resistant derivatives. Deficiencies of FCY1 were not obtained due to the manner in which the screen was implemented. Interestingly, while fcy1∆ mutants were resistant to 5-FC in isolation they were not resistant when co-cultivated together with wild-type, which apparently cross-feed the 5-FU derivative of 5-FC and therefore intoxicate the FCY1-deficient cells. Another interesting finding was the frequent isolation of petite mutants that exhibit 5-FC resistance in spite of their fitness defects in glucose medium. Such mitochondrial deficiencies are known to activate the Pdr3 transcription factor, which induces expression of several ABC transporters and other factors that promote efflux of diverse small molecules. Indeed, the petite mutants were shown to confer decreased retention of a fluorescent dye (i.e. elevated efflux) and also cross resistance to fluconazole (transported by Pdr5, a target of Pdr3). However, there were no follow up experiments demonstrating direct involvement of Pdr3 and Pdr5 in 5-FC resistance of petite mutants. Surprisingly, the mitochondrial deficiencies generally conferred strong cross resistance to Micafungin but not Caspofungin. These echinocandins are important antifungals in clinical settings, so it would be interesting to determine with direct experiments whether Pdr3 is somehow conferring Micafungin resistance in the petite mutants. The manuscript is framed around evolution of drug resistance in clinical settings, but the authors should acknowledge that the findings of this study apply mainly to haploid pathogens and not diploids such as Candida albicans. In summary, the study was interesting, well executed, and well written though a few additional experiments involving Pdr3 would strengthen the final conclusions. Major and minor suggestions for improvement are listed below.

Major concerns

The “predicted alternative path to resistance” involving mitochondria, Pdr3, and Pdr5 that is illustrated in Fig. 1 is very interesting and novel, but not experimentally tested in this study. Do pdr3∆ mutations diminish resistance to 5-FC (and 5-FU) in petite cells as predicted? What about pdr5∆ mutants or any other suspected transporters that regulated by Pdr3?

Minor concerns

1. The study focuses on the evolutionary trajectory of 5-FC resistance in a haploid model organism that is certainly relevant to C. glabrata and other haploid pathogens. But in what ways is this study relevant and not relevant to diploid pathogens that possibly cannot lose mitochondrial function?

2. Cross resistance of petite mutants to fluconazole and micafungin is interesting and relevant to the overall impact of the manuscript. Does Pdr3 contribute to this phenomenon?

3. The analyses of 5-FC resistant isolates that were not petite is confusing. 258 of 276 sequenced genomes contained a mutation in FUR1, but the calls were extremely noisy with over 100 additional mutations in each genome and inconsistency between the calling methods. Sanger sequencing of FUR1 was performed on a large portion of them, but the supplemental tables and the descriptions in the Results do not make it clear which isolate has which mutation (and what sources of evidence support or contradict it). Can a new supplemental table be generated that lists each isolate plus the Sanger sequencing and whole-genome sequencing outcomes (samtools and gatk methods)?

4. The analytical pipeline did not make use of complementation testing, which is gold-standard for identification of causative mutations. In a few places, complementation could resolve and clarify the findings. For example, four resistant isolates did not have mutations in FUR1 and yet behaved like fur1∆ mutants (line 338). Are these mutants complemented by FUR1 on a plasmid, or non-complemented by mating to a fur1∆ strain? What about some of the other ones where the methods were inconsistent or the evidence was incomplete? This data could be added to the new supplemental table proposed above with the overall goals of determining the genetic cause of resistance in each isolate and evaluating the consistency of the methods.

Reviewer #3: Major

I am missing data on the clinical relevance of this study. It would be interesting and important to screen S. cerevisiae clinical isolate for resistance to 5-FC and screen for the mutations identified here (or perhaps novel ones).

I am curious how the authors decided on which antifungal drug concentrations to use. Was this based on some published work? Random? Subinhibitory versus high concentration (in the case of fluconazole)?

Because this manuscript has a ton of data/results, it would be very helpful to have some sort of introduction to both, results and discussion, so the reader can be brought up to speed quickly.

Overall, there needs to be a better description of all the different assays. It is difficult to keep track of all of them. I think some sort of flow chart might work. Alternatively, or in conjuncture, start each method part with a sentence that states the purpose of the experiment might also help.

It is not clear from the methods, which strains were used for the fluctuation analyses? This needs to be clarified. The same goes for parts of the methods.

Fluctuation analyses usually give a rate of (in this case resistance to drug?). I am missing this here.

I am not clear about the reason of generating haploid progeny. When and for which experiment were they used?

The authors identified cross-resistance of rho- mutants to 5-FC and fluconazole and tested efflux capacity. I am guessing that the reason no genomic analysis was done was due to unsuccessful DNA extraction with the method employed? I think genomic data for these mutants would add important information and make this a stronger manuscript. For example, they could look at their genome sequence for mutations in genes known to impact resistance to fluconazole and/or test the expression of genes known to affect fluconazole resistance?

It would be great to have a figure with the FUR1 ORF and the locations of all unique mutations. Along this line, did the authors introduce a few of the major loss of function mutations into the parental strains to confirm the resistance phenotype? Also, in the text it says ‘Supplemental Data 1” for the list of SNPs/indels but they were submitted as suppl 2 excel file, which was a bit confusing.

I am curious if the authors have thought about trying a different selective marker for their fluctuation analysis (Nat or Hyg) and then screen the colonies for 5-FC resistance. Would that have perhaps yielded some Fcy1 mutants?

Discussion, starting at line 479: Can the authors please speculate on what type of unknown mechanism could be responsible and how they would test this? Drug efflux is quite general. In addition, other points made in this part are mostly speculation and I would shorten it a bit.

Minor:

Title of figure 1? Legend is overly long and some details are redundant to what’s described in-text.

Figure S2: I am not sure this figure is necessary.

Line 761: Maybe a reference to Figure S6 would be helpful to visually see what the 96 well plates look like.

Line 779: Please define QC (I am assuming it’s Quality Control?)

Line: 787: which WT strain?

Line 795: what type of centrifuge? 230 g can be all kinds of speed depending on the exact machine.

Figure S3 legend: the sentence ‘A dashed gray line….’ Needs some better wording.

Line 814: Please define SSC and FCS. Not everybody is familiar with flow cytometry

Lines 890-901 should be written in sentence format.

Lines 920-931: This methods may be better positioned before ‘DNA extraction and sequencing.

**Have all data underlying the figures and results presented in the manuscript been provided?**

Reviewer #1: Yes

Reviewer #2: Yes

Reviewer #3: Yes

PLOS authors have the option to publish the peer review history of their article (what does this mean?). If published, this will include your full peer review and any attached files.

Reviewer #1: No

Reviewer #2: No

Reviewer #3: No

---

## [Editor Report · Decision Letter 1]

2 Oct 2023

Dear Dr Durand,

We are pleased to inform you that your manuscript entitled "Cross-feeding affects the target of resistance evolution to an antifungal drug" has been editorially accepted for publication in PLOS Genetics. Congratulations!

Yours sincerely,

Geraldine Butler

Section Editor

PLOS Genetics

Geraldine Butler

Section Editor

PLOS Genetics

Comments from the reviewers (if applicable):

**Data Deposition**

http://datadryad.org/submit?journalID=pgenetics&manu=PGENETICS-D-23-00510R1

**Press Queries**

---

## [Editor Report · Acceptance letter]

16 Oct 2023

PGENETICS-D-23-00510R1 

Cross-feeding affects the target of resistance evolution to an antifungal drug 

Dear Dr Durand, 

We are pleased to inform you that your manuscript entitled "Cross-feeding affects the target of resistance evolution to an antifungal drug" has been formally accepted for publication in PLOS Genetics! Your manuscript is now with our production department and you will be notified of the publication date in due course.

With kind regards,

Anita Estes

PLOS Genetics

On behalf of:
